# Downdip Development of the Ni-Cu-PGE-Bearing Mafic to Ultramafic Uitkomst Complex, Mpumalanga Province, South Africa

**Christoph Gauert [1,2,*] and Armin Zeh [3]**

1 Department of Geology, University of the Free State, Bloemfontein 9300, South Africa
2 Landesamt für Geologie und Bergwesen Sachsen-Anhalt, 06118 Halle, Germany
3 Karlsruher Institut für Technologie, Institut für Angewandte Geowissenschaften—Mineralogie & Petrologie Adenauerring 20b, 76131 Karlsruhe, Germany; armin.zeh@kit.edu
* Correspondence: Christoph.Gauert@sachsen-anhalt.de; Tel.: +49-345-521-2107

**Abstract:** The about 2055-Ma-old mafic to ultramafic Uitkomst Complex in the Mpumalanga Province of South Africa hosts the low-grade-large-tonnage Ni-Cu-PGE deposit, Nkomati. The complex is regarded to represent a satellite to the Bushveld Complex and a feeder to an eroded magmatic reservoir in the southeast. Aeromagnetic surveys and previous drilling indicated an overall northwestern-downdip extension of the complex, but the question is to what extent and in which expression can the complete intrusion be found under cover in the northwest? Answering this, a mineralogical, geochemical and geochronological investigation of a borehole intersection of the whole complex at Little Mamre was carried out, using petrography, XRF, EPMA and LA-ICP-MS U–Pb analyses of zircons for age determination. Although the total thickness of the rock units is larger than to the southeast, emplacement, litho- and mineral chemistry trends, expression of alteration mineralogy and style of sulphide mineralisation are similar. The amount of sulphide mineralisation is on average less than in the southeast. The upper ultramafic unit contains, more frequently, pegmatoidal sections, and the Chromitiferous Harzburgite unit has less massive chromitite layers than the southeastern parts of the complex, whereas the gabbro(-norite) units contain more interstitial liquid with late-stage minerals. The findings confirm that the anvil-shaped intrusion in cross section continues with increased thickness towards northwest at a shallow dip; although approaching the deeper part of the igneous reservoir, mineral compositions are partially more evolved. The overall mineralogical consistency downdip supports a situation of multiple magma replenishment along a flat-lying, northwest–southeast trending conduit, resulting in an evolved cumulus mineral assemblage in the upper part.

**Keywords:** Uitkomst Complex; Nkomati Nickel; South Africa; Ni-Cu-PGE-Co-Cr mineralization; downdip development; Bushveld Complex age

## 1. Introduction

The Nkomati Nickel Mine in the mafic to ultramafic Nkomati Nickel lies in the northeastern part of South Africa near Machadodorp in the Mpumalanga Province. It was the first primary Nickel-producer in South Africa (Figure 1a). The complex hosts a significant orebody of massive to disseminated Ni-Cu-Platinum Group Elements (PGE)(-Co) sulphide mineralisation. Nkomati represents a large Nickel reserve in South Africa hosting measured and indicated mineral resources of 170.25 million tonnes (Mt) of ore grading at 0.35% Nickel and inferred resources of 46.35 Mt grading at 0.40% Ni, in total containing some 784 kt of Nickel metal [1]. In addition, there are measured and indicated Chromium resources of 0.51 Mt grading at about 24.7% $Cr_2O_3$, and proven Chromium reserves of 1.05 Mt at 19.3% $Cr_2O_3$ [1]. After the mining of massive sulphide lenses (MSB mine) on the farm Slaaihoek (SH) from 1995 until 2003, open-pit mining of the predominantly low-grade-large-tonnage Ni-Cu-PGE disseminated sulphide ore body continued until early

in 2021 (Figure 1b). Although resources at the production rates from 2013 to 2020 would last for decades, the Nkomati Ni Mine had to be put in care and maintenance early in 2021 due to low head grades and flotation kinetics distinguishing it from other base metal operations [1]. Mineral alteration and mineralization style influencing the kinetics are reported here.

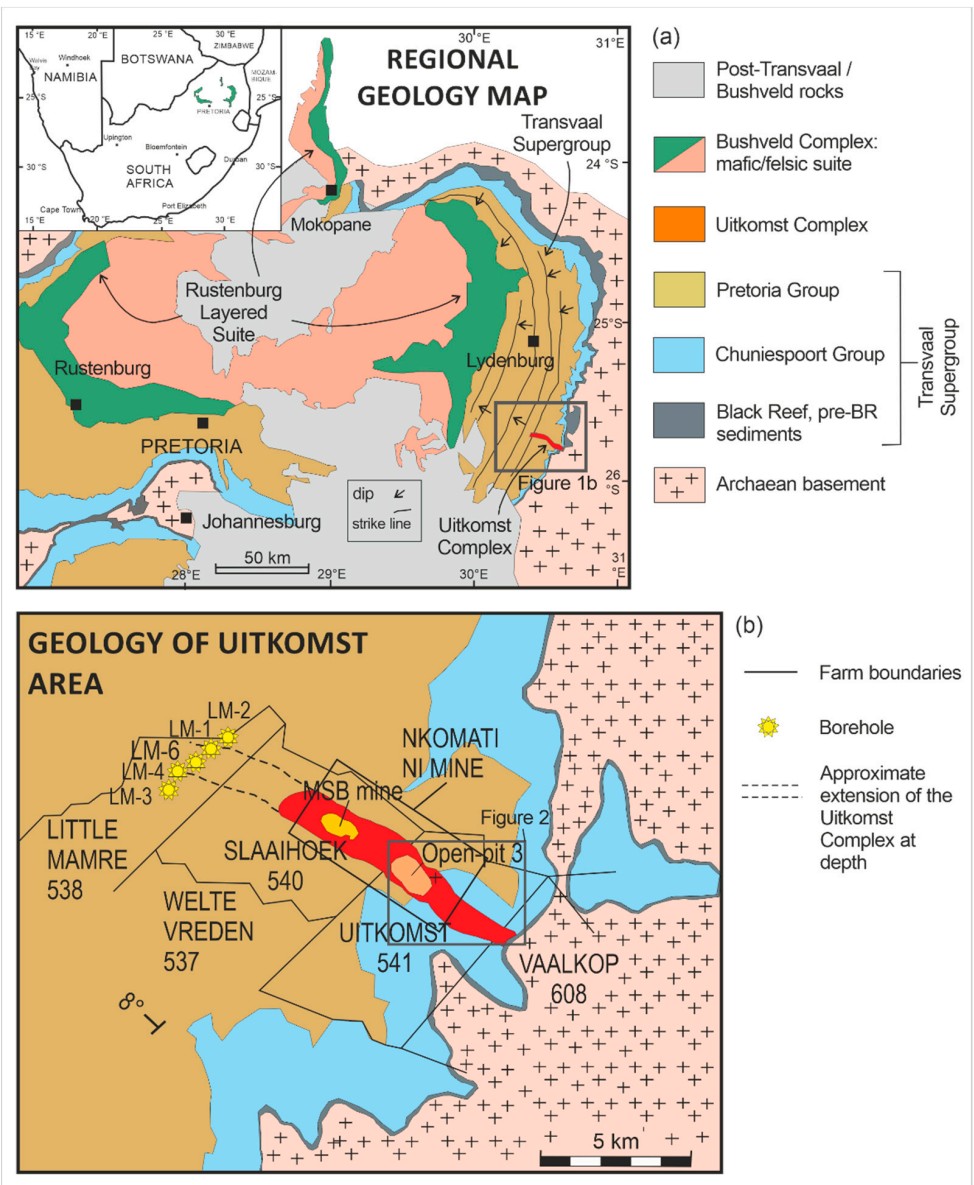

**Figure 1.** Regional geological map of (**a**) the Bushveld Complex, the Transvaal Supergroup basin and (**b**) the Uitkomst Complex with situation of the Nkomati Ni Mine, including 2 shafts at the Massive Sulphide Body (MBS) underground mine and the Open-pit 3 operation. Abbreviations, dip: dip direction, LM: Little Mamre.

The Complex is regarded to be related to the Bushveld magmatic province, possibly representing a satellite and feeder to an eroded magmatic reservoir, due to the northwest-southeast direction of intrusion and the chemical affinity of its chilled margin rocks to marginal Bushveld Complex sills [1]. A magma conduit setting was first reported in 1995 and 1998 [2,3], and later confirmed by others. It is based on the characteristic compositional pattern of the rock units of the complex, indicating magmatic replenishment in an open-system stage, and on the increased width of its contact aureole (Figure 2).

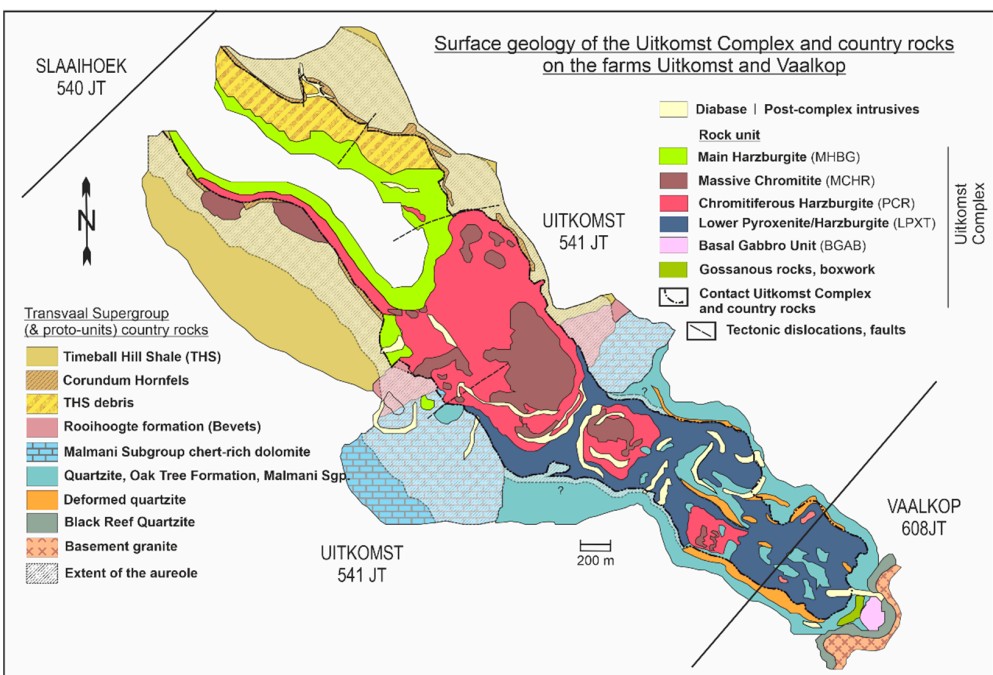

**Figure 2.** Local geological map of the Uitkomst Complex on the farm Uitkomst (UK) with the contact metamorphic aureole (combined map after [4]).

Although open-pit and underground mine development took place mostly on the farms Slaaihoek (SH) and Uitkomst (UK), the northwest-continuation of the complex remains an important emplacement and petrology question. Moreover, a forward projection of the nature and extent of mineralization and alteration of the Chromitiferous Harzburg-ite and Lower Pyroxenite units (Figure 3) that might be expected along the downdip extension of the complex as mining progresses, would give useful insights for future mining.

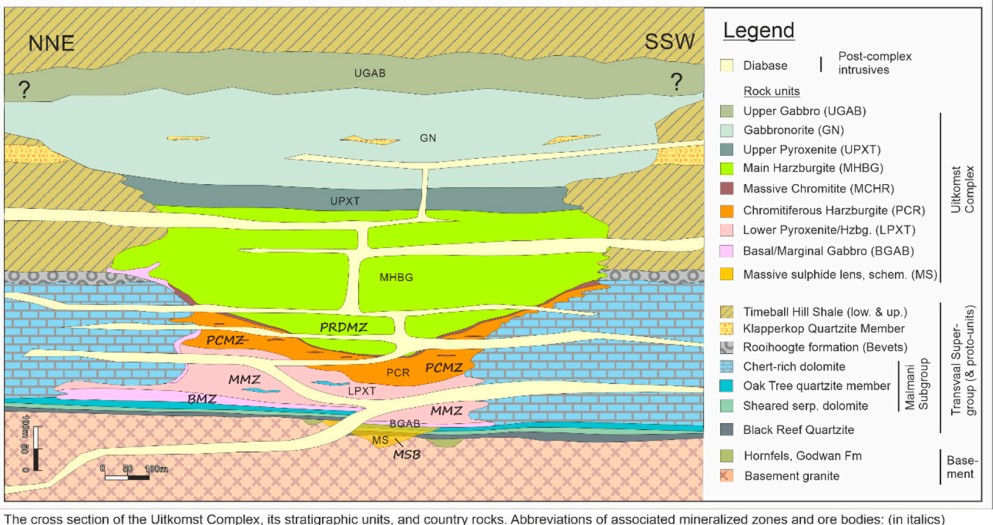

The cross section of the Uitkomst Complex, its stratigraphic units, and country rocks. Abbreviations of associated mineralized zones and ore bodies: (in italics) MSB = Massive Sulphide Body, BMZ = Basal Mineralised Zone, MMZ = Main Mineralised Zone, PCMZ = Chromitiferous Peridotite Mineralised Zone, PRDMZ = Peridotite Mineralised Zone.

**Figure 3.** An idealised schematic cross section of the Uitkomst Complex after [5].

An overall northwestern-downdip extension of the complex had been indicated by aeromagnetic surveys and previous drilling, but only lately an intersection of the entire complex under country rocks was investigated. Borehole LM-6 intersected the complex in its central part on the farm Little Mamre 538, some 14 km northwest of the most southeasterly exposure on the Mpumalanga escarpment (Figures 1 and 4).

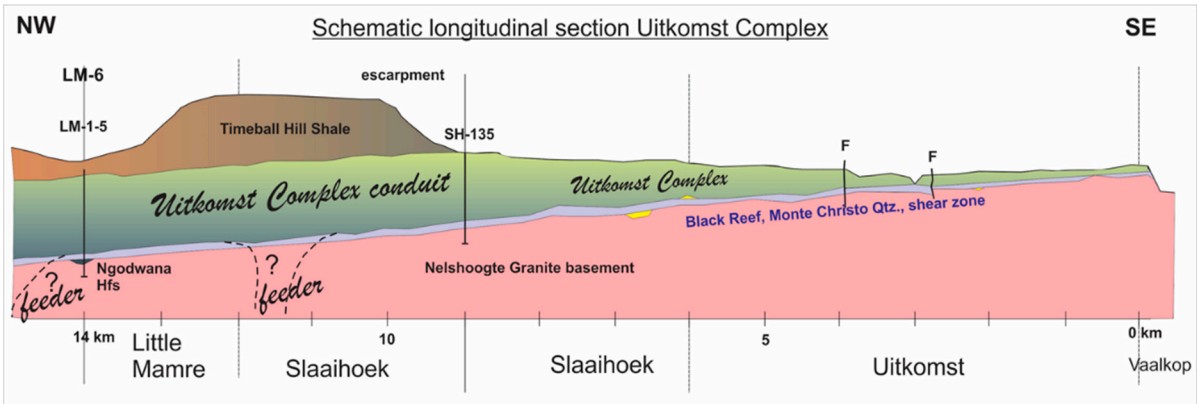

**Figure 4.** Idealised longitudinal section of the Uitkomst Complex from Vaalkop to Little Mamre. F: fault causing vertical displacement, Hfs: hornfels.

The research question and aim of this study is to find out how the mineralogy, the whole rock and mineral chemistry of the Uitkomst Complex, and the thickness of the lithological units continue in their downdip development. Accordingly, it is necessary to characterise the rock sequence at Little Mamre (LM) to answer the questions about emplacement, continuity of the rock units along plunge, mineralogical evolution and alteration processes.

### 1.1. Previous Work

Studies by [2,3,5–22] covered important aspects of the complex' genesis such as age, parent magma composition and magma source, magma emplacement and crystallisation history, origin of the Ni-Cu-PGE-Cr-(Co) ores and mineralisation, and mineral alteration. Studies on contact metamorphism [4,20] still have to be substantiated.

The complex' U–Pb SHRIMP age on baddeleyite of $2044 \pm 8$ Ma [12] was reassessed by an older and slightly more precise multi-grain baddeleyite age [23] of $2054.5 \pm 7$ Ma, indicating a similar age within the margin of error. A high-precision CA-ID-TIMS U–Pb zircon age of $2057.64 \pm 0.69$ Ma [15] shifted the formation point in time to an older age equal to that of the Merensky Reef of the Bushveld Complex. Here we present another LA-ICP-MS U–Pb zircon age of $2055.0 \pm 5.3$ Ma for the downdip lithologies.

An affinity of the magma forming the Uitkomst Complex to the Bushveld primary magma composition B1 [2,3] was supported by mantle-normalised multi-element plots of [15]. Indications of parent magma composition of the complex at LM are presented below.

The postulated dynamic magmatic setting of the complex [10] with repeated magma replenishment was confirmed by [21,22] and [15] based on its cross-section geometry, a reversed fractionation trend in the lower rock units and the compositional homogeneity of the central part of the complex. The magma conduit model is based on the "C- and inverted C-shaped"-compositional pattern of the complex' rock units, represented by the Mg number and the V/Cr ratio (Figure 5). Another argument in favour of a conduit setting is the relatively large amount of sulphide and chromite mineralisation compared to silicate minerals derived from stationary fractional crystallisation/liquid immiscibility in a closed magmatic system [10].

A predominantly magmatic sulphur source however with significant, but variable shale- and dolomite-derived sulphur assimilation by the magma was confirmed by several studies [8,15,16,20]. These authors suggested that the amount of assimilated shale-hosted sulphur with strongly negative $\delta^{34}$S ratios could have been reduced due to S loss during contact metamorphism.

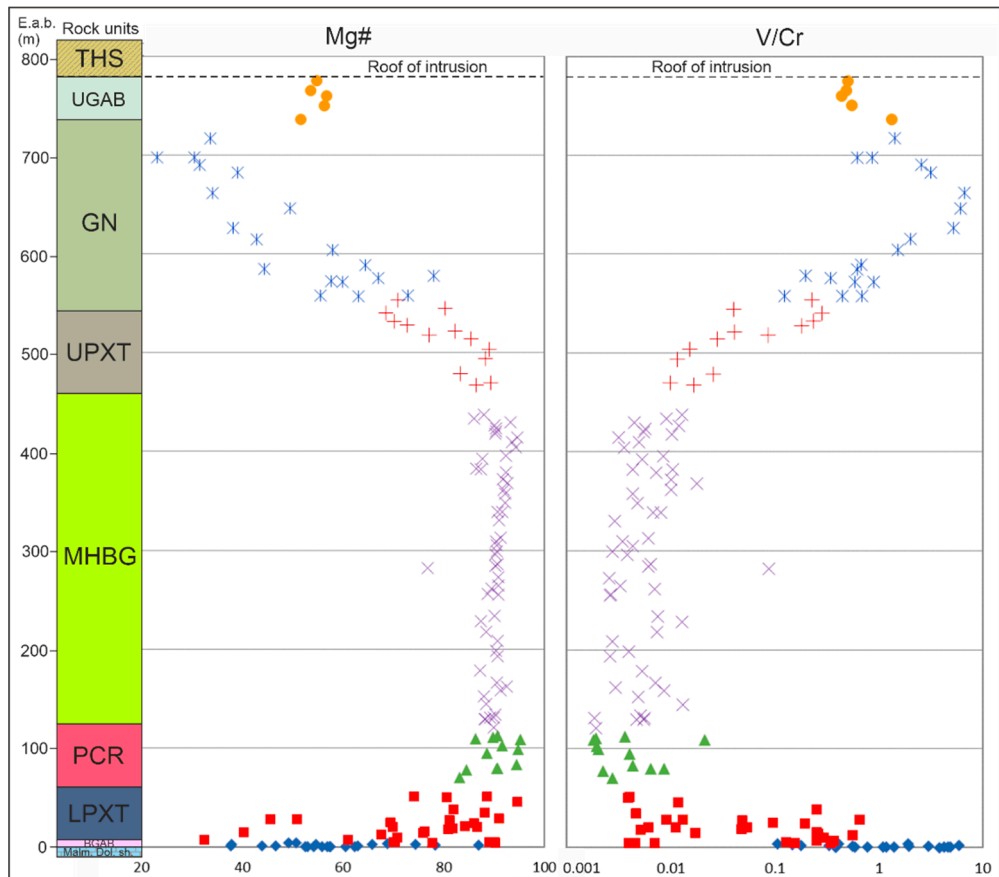

**Figure 5.** Geochemical and mineral chemistry composite profile of the Uitkomst Complex at LM (LM-1 and LM-4), SH and UK [3]. BGAB: Basal Gabbro, LPXT: Lower Pyroxenite, PCR: Chromitiferous Harzburgite, MHBG: Main Harzburgite, UPXT: Upper Pyroxenite, GN: Gabbronorite, UGAB: Upper Gabbro, Malm.Dol.sh.: sheared Malmani Dolomite, THS: Timeball Hill Shale, E.a.b.: Elevation above base, Mg#: $Mg*100/(Mg + Fe^{2+})$ in atomic proportions.

An extensive contact aureole consisting of meta-carbonate, metapelite and quarzite up to several hundred meters wide appears to be too large for an intrusion of that size ([4] and Figure 2; Appendix A, Figure A1: Plate 1a,b). This supports the existence of a replenished conduit, through which a magma volume of ten to hundred times may have pulsed.

Although the uppermost part of the complex in drill holes at LM has been described previously [3], the innovative part of this article is the recognition of the first complete vertical profile of the complex at this position and its implications for the downdip development. In order to assess the northwestern-downdip extension of this magmatic conduit, this study investigates the recently available LM-6 drill core from the farm LM, which are compared to that of drill cores LM-1 and -4. Important questions comprise the continuity of mineralogy and geochemistry of the layered igneous succession, whether the supposed magmatic conduit changes to a subvertical feeder, and if the intensity of mineral alteration decreases under country rock cover.

## 1.2. Geological Setting of the Uitkomst Complex

Situated about 200 km east of Pretoria (Figure 1) and some 65 km southeast from the nearest Bushveld rocks along the eastern Mpumalanga Province escarpment of South Africa, the magma forming the Uitkomst Complex intruded along a craton-wide crustal lineament from NW to SE into the lower three lithological units of the Transvaal Supergroup, the Black Reef quartzite, the Malmani Dolomite and the Timeball Hill Shale (Figures 1, 2 and 4). Exposures exist on the farms SH, UK and Vaalkop over a distance of approximately 9 km. The plunge direction of the complex on the farms SH and LM shifts slightly to the west,

possibly caused by gradual displacement of several SW-NE trending strike slip faults or by a bending of the intrusive direction (Figure 1b). Based on geochemical and mineralogical evidence, it was suggested that the intrusion represents a satellite body of the Bushveld Complex feeders [2,16,20].

The shape of the intrusion changes depending on elevation above base from a lower trough to an upper funnel shape of the upper rock units (Figure 3). Its width therefore ranges between 750 and 1600 m. The rock units of the complex are from bottom to top: Basal Gabbro unit, Lower Pyroxenite unit, Chromitiferous Harzburgite unit with a massive chromitite layer on the top, Main Harzburgite unit, Upper Pyroxenite unit, Gabbronorite unit and Upper Gabbro unit. The complex has basal, upper and partly lateral chilled margin rocks (Figure 3).

Initially, the primary igneous mineral assemblage and the magmatic processes forming the ore were of interest. However, looking at the buried igneous suite underneath the Transvaal sediment cover, still a strong degree of magmatic-hydrothermal alteration during solidification is obvious, indicating strong activities of probably late magmatic fluids. It has previously been shown that magmatic and metamorphic waters are responsible for the alteration [20].

### 1.3. Downdip Development of the Complex

The known downdip extension of the Uitkomst Complex is 14 km to the northwest. An idealised longitudinal section of the Uitkomst Complex from Vaalkop to LM (Figure 4) depicts an undulating floor contact along a shear zone within the upper Monte Christo quartzite of the Malmani Dolomite Subgroup, as indicated by drill holes on the farm SH and projected towards LM. The intrusion emplaced bedding parallel over its entire known length, preserving an almost identical angle of dip of igneous rocks as the Transvaal Supergroup strata. It is of importance for exploitation of the orebody and mine development to know the continuity of the shallowly plunging complex, or the position where it may change into a sub-vertical dyke. A feeder dyke to the complex could so far not be established. The position of LM-6 in the LM drill hole line, the only borehole vertically intersecting the entire complex in this area, is shown in Figure 6.

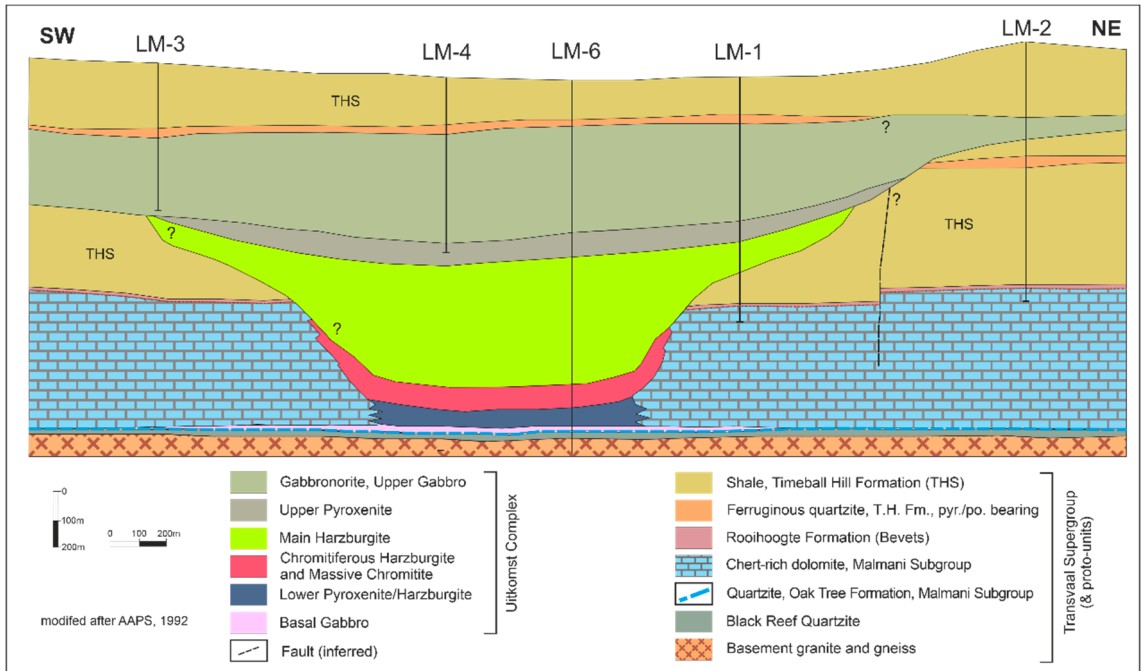

**Figure 6.** Approximate position of LM-6 in the SW-NE drill hole LM-1 to -4 line, the only borehole intersection of the entire complex on the farm LM and the inferred shape of the complex there; Abbreviations: pyr./po. = pyrite/pyrrhotite.

## 2. Materials and Methods

In total, 123 rock specimens of different lithotypes from drill core LM-6 were sampled in regular distances. The main focus of the investigation was to characterize an entire profile of the rock units at LM. Polished thin sections of all rock samples were produced for petrography and mineral chemistry. Special focus was laid on pegmatoidal rock types. For the microscopic identification of the minerals an optical BH-2 microscope from Olympus was used. All transparent minerals in thin section were identified by transmitted light, except the ore minerals, which were investigated under reflected light.

Apatite, zircon, titanite, chromite, pentlandite, pyrrhotite and chalcopyrite were analysed in about 13 thin sections using a Jeol Superprobe Analyser at Rhodes University, Grahamstown, South Africa, under analytical conditions of 20 kV, $0.2 \times 10^{-7}$ A, with counting times of 20 s on peaks and 10 s on backgrounds, using Bushveld chromite, pure V-, Ti-, Mn-, Ni-oxide, synthetic glass for Zn, plagioclase feldspar and pure quartz as standards.

Whole rock pulps were provided for whole rock geochemistry. The X-ray fluorescence (XRF, Malvern Panalytical, Bloemfontein, South Africa) measurements were carried out with the Panalytical Axios XRF spectrometer and interpreted and quantified using the Super Q analytical software (Version 5x, Malvern Panalytical, Almelo, The Netherlands). Loss on ignition, adhesive and crystal water were determined in a muffle oven at 100 and 1000 °C. Results within a total interval 98.5 to 101 wt.% were included.

### LA-ICP-MS U–Th–Pb Dating

Zircons used for U–Pb isotope age determination were separated from sample LM-6-7 of the Lower Pyroxenite unit. All dated zircons were selected from heavy mineral concentrates derived from a bulk rock sample. The zircon grains were mounted (ca. 25 grains) in epoxy resin and ground and polished to expose the crystal cores. Internal zoning patterns were characterised by cathodoluminescence (CL) and back scattered electron (BSE) imaging using a Gatan mini-CL detector coupled to a JEOL JSM 6490 raster electron microscope at Goethe University Frankfurt (GUF). Subsequently, uranium, thorium and lead isotopes were analysed using a ThermoScientific Element 2 sector field ICP-MS coupled to a Resolution M-50 (Resonetics) 193 nm ArF excimer laser (ComPexPro 102F, Coherent) system at Goethe University Frankfurt. Data was acquired in time resolved-peak jumping–pulse counting/analogue mode over 466 mass scans, with a 20 s background measurement followed by 21 s sample ablation. Laser spot-size was 30 µm for unknowns and the standard zircon GJ1 (primary standard), Plesoviče, and two previously CA-ID-TIMS dated zircons of the Bushveld Complex (PGMT and CH12-3), and all three were used as secondary standards. The results of the standard zircon measurements are shown in Table S1 of the Supplementary Materials. Sample surface was cleaned directly before each analysis by three pulses pre-ablation. Ablations were performed in a 0.6 L min$^{-1}$ He stream, which was mixed directly after the ablation cell with 0.007 L min$^{-1}$ $N_2$ and 0.83 L min$^{-1}$ Ar prior to introduction into the Ar plasma of the SF-ICP-MS. All gases had a purity of >99.999% and no homogeniser was used while mixing the gases to prevent smoothing of the signal. Signal was tuned for maximum sensitivity for Pb and U while keeping oxide production, monitored as $^{254}UO/^{238}U$, below 0.4%. The sensitivity achieved was in the range of 12,000 cps/µg g$^{-1}$ for $^{238}U$ with a 33 µm spot size, at 5.5 Hz and 5 J cm$^{-2}$ laser energy. The typical penetration depth was about 15 µm. The two-volume ablation cell (Laurin Technic, Australia) of the M50 enables detection and sequential sampling of heterogeneous grains (e.g., growth zones) during time resolved data acquisition, due to its quick response time of <1 s (time until maximum signal strength was achieved) and wash-out (<99.9% of previous signal) time of about 2 s. Raw data was corrected offline for background signal, common Pb, laser induced elemental fractionation, instrumental mass discrimination, and time-dependent elemental fractionation of Pb/U using an in-house MS Excel$^{©}$ spreadsheet program [24,25], with modifications explained in [26]. A common-Pb correction based on the interference- and background-corrected $^{204}Pb$ signal and a model Pb composition [27] was carried out. The $^{204}Pb$ content for each ratio was estimated by subtracting the average

mass 204 signal, obtained during the 20 s baseline acquisition, which mostly results from $^{204}$Hg in the carrier gas (c. 300 cps), from the mass 204 signal of the respective ratio. For the analysed sample, the calculated common $^{206}$Pb contents was mostly <2% of the total $^{206}$Pb but in some cases exceeded 5% (Supplementary Material, Table S2). Laser-induced elemental fractionation and instrumental mass discrimination were corrected by normalisation to the reference zircon GJ-1 [28]. Prior to this normalisation, the inter-elemental fractionation ($^{206}$Pb*/$^{238}$U) during the 21s of sample ablation was corrected for each individual analysis. The correction was done by applying a linear regression through all measured, common Pb corrected ratios, excluding the outliers ($\pm$2 standard deviation; 2 SD), and using the intercept with the y-axis as the initial ratio. The total offset of the measured drift-corrected $^{206}$Pb*/$^{238}$U ratio from the "true" ID-TIMS value (0.0983 $\pm$ 0.0004; ID-TIMS GUF value) of the analysed GJ-1 grain was about 4–6% during the analytical sessions.

Reported uncertainties (2$\sigma$) of the $^{206}$Pb/$^{238}$U ratio were propagated by quadratic addition of the external reproducibility (2 SD) obtained from the standard zircon GJ-1 and the within-run precision of each analysis (2 SE; standard error). In case of the $^{207}$Pb/$^{206}$Pb we used a $^{207}$Pb signal dependent uncertainty propagation (see [25]). The $^{207}$Pb/$^{235}$U ratio is derived from the normalised and error propagated $^{207}$Pb/$^{206}$Pb* and $^{206}$Pb*/$^{238}$U ratios assuming a $^{238}$U/$^{235}$U natural abundance ratio of 137.88 and the uncertainty derived by quadratic addition of the propagated uncertainties of both ratios. The analytical results are presented in the Supplementary Materials. The accuracy of the method was verified by analyses of reference zircon Plesovice, PGMT and CH12-3, which yielded Concordia ages of 337.8 $\pm$ 0.9 Ma (*n* = 27), 2057.1 $\pm$ 5.1 (*n* = 15) and 2055.3 $\pm$ 4.1 (*n* = 33), respectively. These ages are within error identical to the quoted TIMS values of 337.31 $\pm$ 0.37 Ma for Plesoviče zircon [29], and CA-ID-TIMS values of 2055.40 $\pm$ 0.30 Ma and 2055.68 $\pm$ 0.29 Ma for PGMT and CH12-3 [30], respectively. The analytical results of standards and unknowns are presented in Section 3.5 below and/or in Table S2 of the Supplementary Materials. The data was plotted using the software ISOPLOT (Version v. 4.15, Berkeley Geochronology Centre, Berkeley, CA, USA) [31].

## 3. Results

### 3.1. Lithologies and Petrography of the Complex at LM-6

In line with earlier assumptions, the intrusion dips with a similar angle to the country rock (Figure 4), with an undulate floor contact on sheared Malmani quartzite on top of a thick Black Reef quartzite layer. Underground exposure in the MSB mine at SH and detailed drill hole floor contact intersections on UK confirm its undulation and partially vertical displacement, projecting this information to the LM side. The drill hole line on farm LM (Figure 6) implies the proximal presence of dolomite to the northeast and the possibility of magma-country rock interaction. Detailed petrography allowed to estimate the modal proportions of the entire profile of the Uitkomst Complex as intersected in LM-6, attributing the alteration mineral proportions to the respective primary igneous minerals (Figure 7).

A thin bottom chill zone of aphanitic fine-grained ophitic, altered and veined basal gabbro is overlain by at least 3 m of phaneritic, mineralised gabbroic orthocumulate, representing a first magma pulse of basaltic composition along a shear zone. The Basal Gabbro unit varies in thickness between 0 and 15 m throughout the entire complex with an average of 5.6 m [3], and continues in similar thickness in LM-6 at LM. The degree of saussuritisation and uralitisation increases in the upper part, and so does the magnetite content (Supplementary Materials, Table S3).

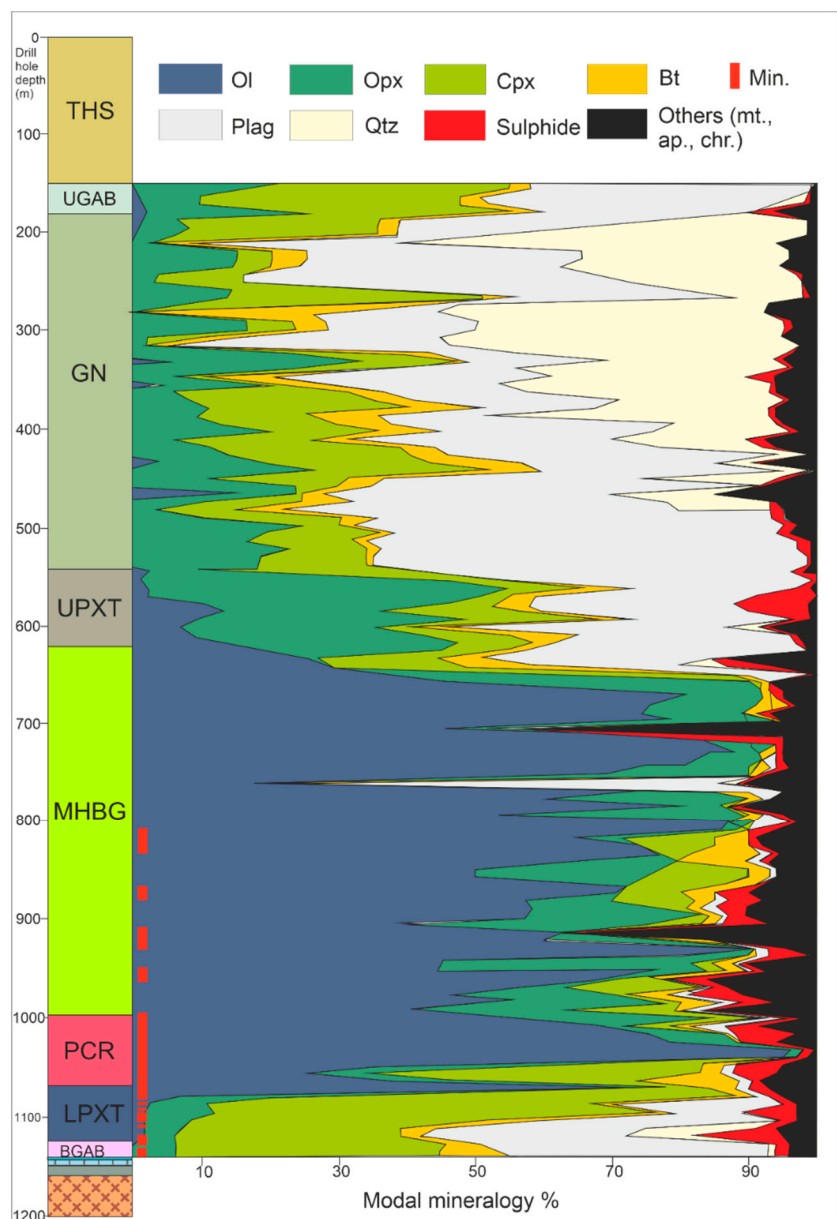

**Figure 7.** Modal proportions of samples of all rock units of the Uitkomst Complex in borehole LM-6. Abbreviations: Rock units as in Figure 5, Ol = olivine, Opx = orthopyroxene, Cpx = clinopyroxene, Bt = biotite, Plag = plagioclase, Qtz = quartz, Sulphide = base metal sulphides, mt = magnetite, ap = apatite, chr = chromite, Min. = less than 5% base metal sulphide mineralisation, in the Basal Gabbro unit ~10%, indicated by red lines in the lower left of the diagram.

The ortho- to adcumulates of the Lower Pyroxenite unit in LM-6 are thicker than the unit in the rest of the complex (about 63 m in LM-6 compared to about 50 m on average). A broad spectrum of alternating rock types, including feldspar-bearing lherzolite, sulphide-rich feldspathic wehrlite, amphibolite, poikilitic harzburgite and pegmatoidal pyroxenite with calc-silicate xenoliths of maximal 4 m thickness is observed. Large portions of the primary minerals are altered to amphibole, serpentine to talc-carbonate and phlogopite (Supplementary Materials, Table S3). The Lower Pyroxenite unit contains the bulk of the disseminated pyrrhotite–pentlandite–chalcopyrite mineralisation (Appendix A: Figure A3, Plate 3a). The frequent occurrence of primary and secondary magnetite is conspicuous (Supplementary Materials, Table S3).

Similar to sections in the southeast, the mineralisation in the LM-6 intersection consists of interstitial composite sulphide aggregates showing complex intergrowth of pyrrhotite, and to a lesser extent chalcopyrite and pentlandite, with various silicate minerals and (titano-)magnetite along the crystal contacts (Appendix A: Figure A3, Plate 3a,b). Granular pentlandite is situated along the grain boundary between pyrrhotite and silicate minerals. Significant overgrowth of the pyrrhotite and pentlandite by bladed, secondary amphibole has taken place causing an increase in the complexity of the pentlandite grain boundaries (Appendix A: Figure A3, Plate 3c). Various accessory secondary oxide and sulphide minerals occur (Supplementary Materials, Table S3).

The Chromitiferous Harzburgite unit in LM-6 with about 74 m thickness lies also above the average thickness of the unit (about 60 m) at SH and UK. The contact of the Chromitiferous Harzburgite with the underlying Lower Pyroxenite unit is gradational. The unit is a strongly to entirely altered, mineralised, foliated and veined ad- to orthocumulate of chromite-bearing medium-grained harzburgite to dunite (Appendix A, Figure A4, Plate 4d). Only thin massive chromitite layers (MCHR) are found compared the thicker ones on UK and SH. Because of the pervasive alteration, the magmatic minerals were replaced by talc, carbonate, phlogopite, serpentine, amphibole and resulted in two strongly altered rock types: a talc-amphibole-carbonate-chlorite rock and a serpentinite (Appendix A, Figure A2, Plate 2a). Chromite proportions vary in a matrix of completely altered harzburgite. Disseminated pyrrhotite–pentlandite aggregates and magnetite veins occur. The MCHR further decreases in thickness from SH towards the northwest, and is reduced to several layers with less than 1.7 m thickness at LM.

Salient features of the alteration mineralogy of the cumulate rocks are the increased amount of amphibole and muscovite in the Basal Gabbro unit, the mostly amphibolitised pyroxenes of the Lower Pyroxenite unit, and the serpentine-talc-carbonate-chlorite dominance in the Chromitiferous Harzburgite unit. However, the Chromitiferous Harzburgite unit shows less altered portions of silicates at the transition to the Main Harzburgite unit. Hydrothermal, partly deuteric alteration is widespread in the complex, but pronounced in its lower three and upper two units.

The Main Harzburgite unit in LM-6 exceeds the average thickness of 330 m of the unit in the southeastern part of the complex, representing more than one third of the total thickness of Uitkomst Complex. The unit is a continuous sequence of harzburgite in the form of an olivine-orthopyroxene meso- to adcumulate (Appendix A: Figure A2, Plate 2b). In some places, the harzburgite grades over short distances into thicker dunite portions, and frequently it contains minor intercumulus plagioclase. The unit shows a weak meter-scale layering which is caused by variations in grain size and composition. Net-textured sulphide mineralisation is strongest in the lower 50 m of the Main Harzburgite unit. The main alteration type of the unit is serpentinisation, ranging from weakly altered parts to local occurrences of serpentinites. An approximately 20 m thick, partly pegmatitic layer in the upper Main Harzburgite (Figure 7) may indicate the transition into the Upper Pyroxenite Unit.

The Upper Pyroxenite unit consists usually of three subunits, a lower olivine–orthopyroxenite, an orthopyroxenite and an upper norite to gabbronorite orthocumulate and mesocumulate, and is up to 60 m thick. In LM-6, the unit is about 70 m thick, including a substantial moderately serpentinised olivine–orthopyroxenite section.

The Gabbronorite and Upper Gabbro units together form the uppermost sequence of the Uitkomst Complex with an average thickness of 250 m. In LM-6, it exceeds 300 m thickness. At LM, the Gabbronorite unit is characterised as a laterally extensive sill-like body [2] and can be subdivided into a main Gabbronorite unit and an Upper Gabbro unit [20] (Figure 3). The sub-rock types range from a basal norite through gabbro to diorite in the central upper part [12,16] (Figure 7). The petrography of the Gabbronorite unit in drill core sections of LM-1 and LM-4 broadly corresponds to the findings of the Gabbronorite unit in LM-6, except for frequent occurrence of Mg-spinel and less titanite in LM-1/-4 (titanite in LM-6: Appendix A, Figure A4, Plate 4b,c).

The lower part of the Gabbronorite unit is relatively unaltered and consists of norite, gabbronorite and olivine gabbro. The upper Gabbronorite and Upper Gabbro units are strongly altered with a secondary mineral assemblage consisting of saussuritised plagioclase, chlorite, biotite and tremolite [14,20] (Appendix A: Figure A2, Plate 2c,d; Figure 4, Plate 4a). The unit is also strongly contaminated with quartz by assimilation of country rocks such as the Klapperkop quartzite member of the Timeball Hill Shale (Figure 3).

In LM-6, the Lower Pyroxenite and the Upper Pyroxenite unit are less thick than in other positions of the complex, whereas the Main Harzburgite, Gabbronorite and Upper Gabbro units are thicker. Overall, the intersection of the downdip extension of the complex in LM-6 shows a larger thickness of about 885 m (986 m intersection with about 100 m postintrusive dolerite/diabase) compared to an estimated average thickness of 750 m (about 850 m intersections with about 100 m diabase intrusions, [2]).

### 3.2. Geochemical and Mineral Chemistry Trends in LM-6

The geochemical profile, as determined by XRF analyses, of LM-6 shows an inverted 'C'-shaped trend for Mg#, Cr and olivine-corrected Ni/Cu ratio (Figure 8a), whereas the $TiO_2$, V/Cr ratio, Rb, Zr and Ba show a 'C'-shape (Figure 8b), similar to the composite geochemical profile from the farms UK, SH and LM [3]. The coloured symbols represent the different rock units as depicted in the stratigraphic column. Compatible elements increase from the base of the Lower Pyroxenite and are constantly high up to about 500 m elevation above base, where a steady decrease to lowest values at the top of the Gabbronorite unit begins. The upper 250 to 300 m of the complex in contrast show a steady increase of incompatible elements. However, in all rock units are intervals of higher geochemical variability, indicating that careful comparisons are still necessary. Incompatible trace element ratios such as Zr/Rb and Zr/Y have a bipartite distribution pattern, in that these are relatively low in the pyroxenitic to peridotitic units, and up to a magnitude higher in concentration in the gabbro(norit-)ic units. This is probably more obvious at LM compared to the profile at SH and UK, indicating larger source compositional and differentiation variations between the units downdip.

In detail, the Basal Gabbro unit is geochemically characterised by an average Mg# of 46.5 (range of 41.64 to 50.9), and average ratios of Cr/V of 0.3 (0.1 to 0.5), and of Ni/Cu of 0.5 (0.2 to 1.0), as well as Zr values of 160 to 180 ppm and Ba values of 250 to 400 ppm (Figure 8a,b). The Lower Pyroxenite unit in contrast is characterised by an average Mg# of 75.3 (69.9 to 82.8), and average ratios of Cr/V of 7.4 (2.2 to 19.0), and of Ni/Cu of 4.9 (1.8 to 13.0), as well as Zr values of 40–80 ppm, in the Main Harzburgite below 30 ppm (Figure 8a,b). Similar to the geochemical profile at UK [3], the highest average Mg# in LM-6 are found in the MHZB unit with 86.5 (71.2 to 90.6), as well as with a maximum ratio of Cr/V of 209.3 (34.4 to 596.1), and of Ni/Cu of 202.6 (2.0 to 5421.3). The highest $TiO_2$ (max. of about 4 wt.%), Zr (max. 400 ppm) and Ba (max. 800 ppm) values are found in the Upper Gabbro unit (Figure 8b).

The transition from Basal Gabbro unit into Lower Pyroxenite is chemically characterised by distinctly upwards rising Mg#, Cr/V and Ni/Cu ratios, and a strong decrease in Zr, Ba and $TiO_2$ content (Figure 8). Geochemically, the increase of Mg# from the Lower Pyroxenite (minimum 69.9) to the Chromitiferous Harzburgite unit (max. 92.7), of Cr/V (2.2 to max. 596) and of Ni (ol. corr.)/Cu ratios (2.0 to 1000) are significant. This, together with further decreases in Zr, Ba and $TiO_2$ content (Figure 8), may indicate the onset of inflow of ultramafic magma surges in the centre of the intrusion.

Compositional variations of the melt caused by differentiation are reflected by variable olivine-corrected Ni/Cu ratios, which are relatively high for the pyroxenitic to peridotitic units and up to two magnitudes lower for the gabbro(norit)ic units.

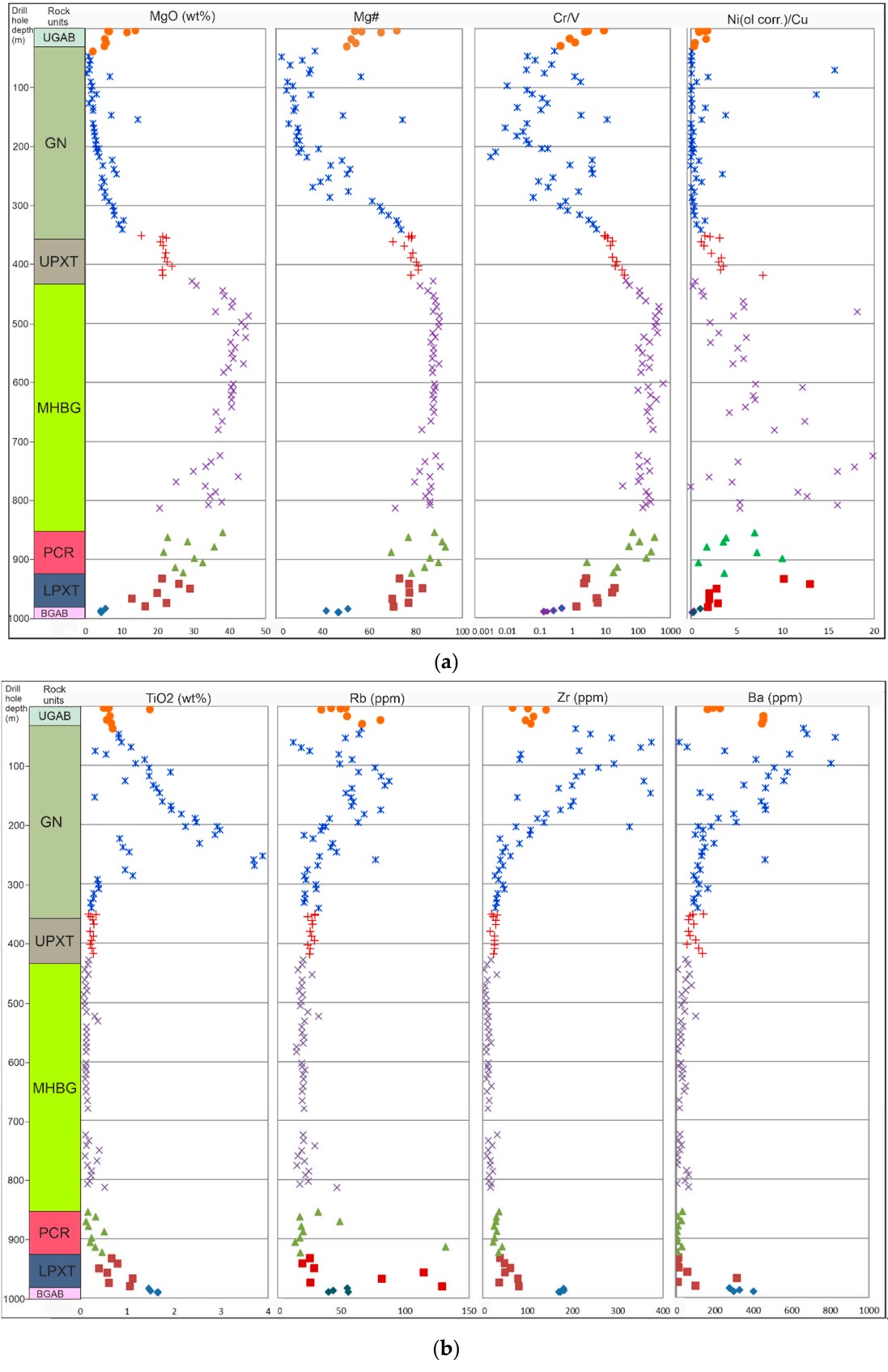

**Figure 8.** Overall geochemical XRF profile of element(oxide)s and ratios in LM-6. Abbreviations: Rock units as in Figure 5. (**a**) compatible element(oxide) ratios; Mg#: Mg*100/(Mg + Fe$^{2+}$) in atomic proportions, Ni (ol. corr.)/Cu: For olivine-corrected Nickel/Copper ratio (dimensionless). (**b**) incompatible element(oxide) ratios (dimensionless).

The similarity of major and trace element chemistry between the Basal Gabbro samples, both chills and phaneritic varieties and the B1 micropyroxenite, thought to represent the parental magma to the Bushveld Complex [32], is illustrated in B1-normalised spiderdiagrams of selected trace element(-ratio)s (Figure 9a). The average phaneritic Basal Gabbro unit of LM-6 is slightly enriched in Ni and strongly in Cu, a result of its sulphide content, while the B1 magma is significantly richer in Cr, relative to the Basal Gabbro unit on the farm Uitkomst.

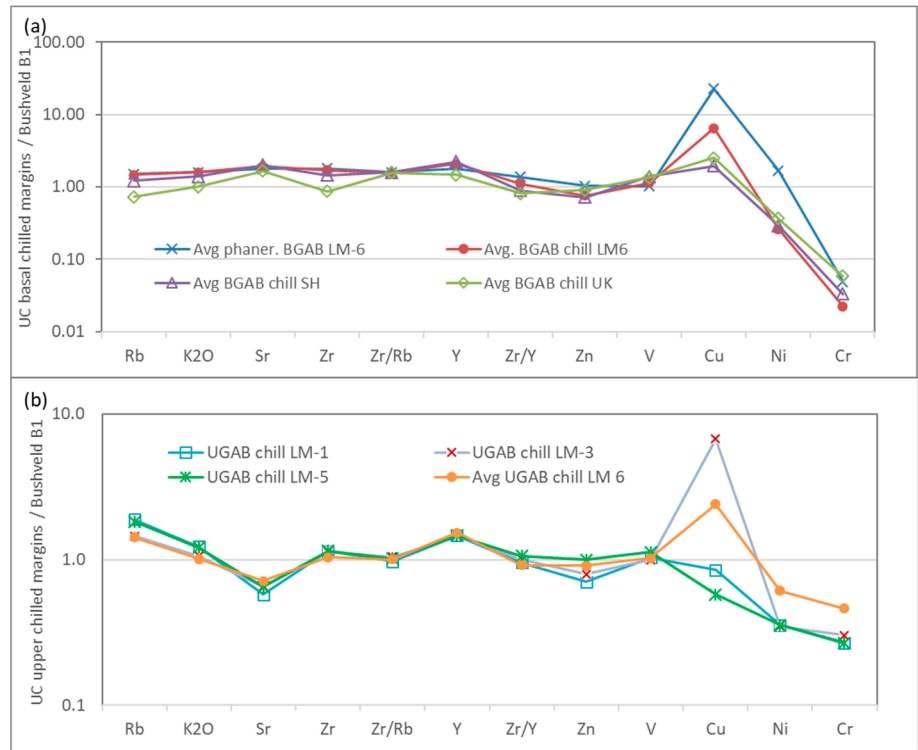

**Figure 9.** Minor element concentrations and ratios of rocks from (**a**) the Basal Gabbro unit at UK, SH and LM, and (**b**) the Upper Gabbro units of the UC at LM, normalised to Bushveld B1 micropyroxenite of [32].

The Upper Gabbro chill composition at LM-6 and in the other LM drill cores are generally similar to B1 in their overall trace element pattern (Figure 9b), except for the enrichment in Cu and depletion in Ni and Cr. Interestingly, the Cr content of the average LM-6 upper chill lies at half of the B1 composition, and that of the other LM Upper Gabbro samples at a third of B1, indicating a more evolved source for the top chill zone.

Since the main mineral compositional trends at SH and UK mimick the whole rock compositional patterns [1,3], this study does not focus on the composition of the cumulus minerals, but rather on that of common accessory minerals such as titanite, ilmenite, apatite, chromite, amphibole, biotite and albite, especially in pegmatite 'pockets' within the Main Harzburgite unit and in the dioritic part of the GN unit.

Additionally, the chemical composition of chromites of the lower to central rock units gives valuable hints for the differentiation process in the magma conduit. Apatite, titanite and chromite EPMA data shows considerable compositional variation in the LM sections (Supplementary Material, Table S2). A chemical characterisation of apatite over the stratigraphy shows constant $P_2O_5$ (avg. $43.3 \pm 0.82$ wt.%) and CaO (avg. $52.98 \pm 0.99$ wt.%) compositions, but variable minor F (1.06 to 4.26 wt.%), Cl (0.02 to 3.16 wt.%) and FeO (0 to 0.85 wt.%) contents (Figure 10), as well as traces of K, Si, Na, Al, Fe and Ba (Supplementary Material, Table S2).

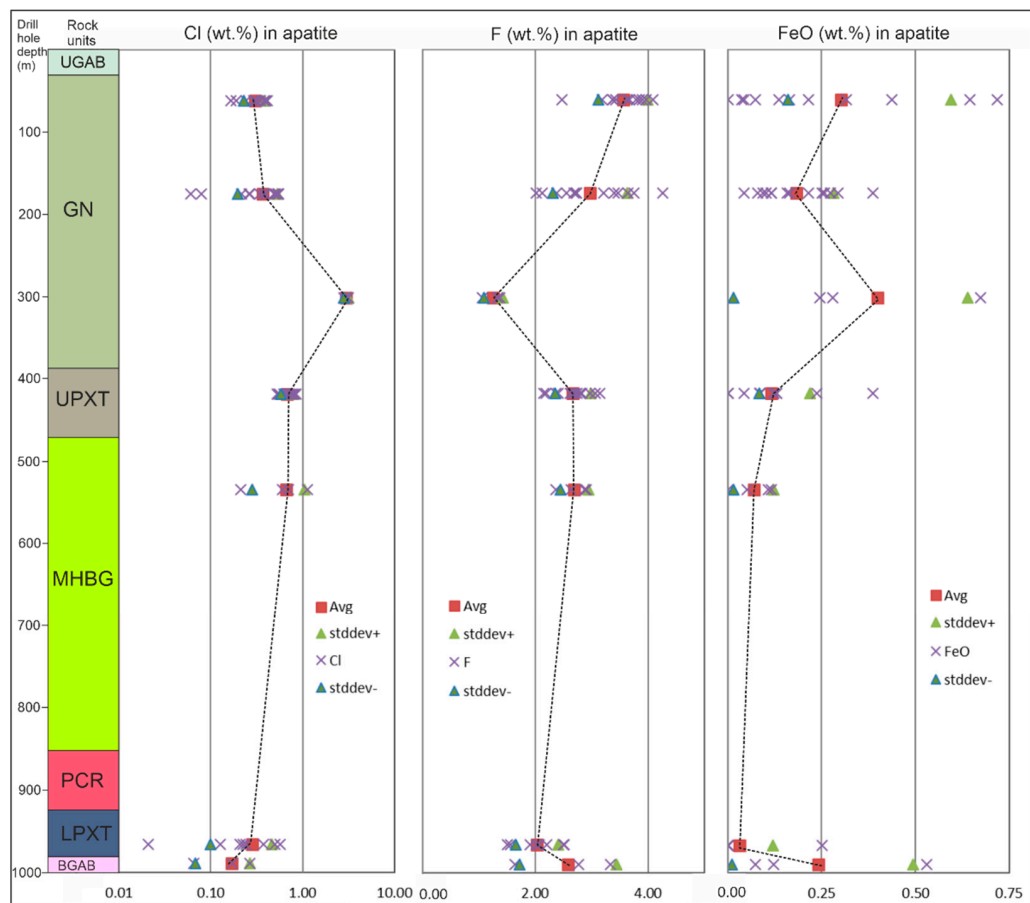

**Figure 10.** Stratigraphic variation of trace elements (EPMA) in apatite. Abbreviations: Rock units as in Figure 5. Avg: average value, Stddev+/−: average plus/minus standard deviation.

Based on limited data so far, the FeO content in apatite increases from the Lower Pyroxenite unit slightly upwards, and the F content in apatite also increases, whereas Cl shows an inverted trend in the Gabbronorite unit. The apatite trace element trends follow the trend of incompatible trace elements in whole rock analyses.

Late stage apatite is commonly associated with interstitial quartz both in the lower as well as upper gabbro. In pegmatoidal sections of the ultramafic rocks it occurs together with large biotite porphyroblasts, green amphibole and occasional carbonate veins.

Zircons in LM-6 vary in composition, but contain more than 200 ppm of $P_2O_5$ on average, and between 0.17 and 1.38 wt.% of $HfO_2$ (Figure 11), as well as traces of Al, Ti, Mn, Fe and REE (Supplementary Material, Table S2).

The titanites in LM-6 contain as trace elements Al, Fe, Ta, and V, some crystals contain Mn, Mg, Na, Zr, F and some REE (Supplementary Material, Table S2); based on limited data, the increase especially of the $Ta_2O_5$ content in the titanites with stratigraphic height from below detection to 1.8 wt.% is noticeable (Figure 11).

The trace element chemistry of titanite in evolved amphibole-bearing gabbroic rocks constitutes a tool for understanding late-stage igneous and metasomatic processes. Therefore, LM titanites in textural relationship with adjacent plagioclase, Fe-Ti oxides, and mafic minerals were determined to have detectable to elevated concentration of the oxides of Nd, Zr, Ce, Nb, Ta, and V ranging from tenth to several wt.% (Supplementary Material, Table S2).

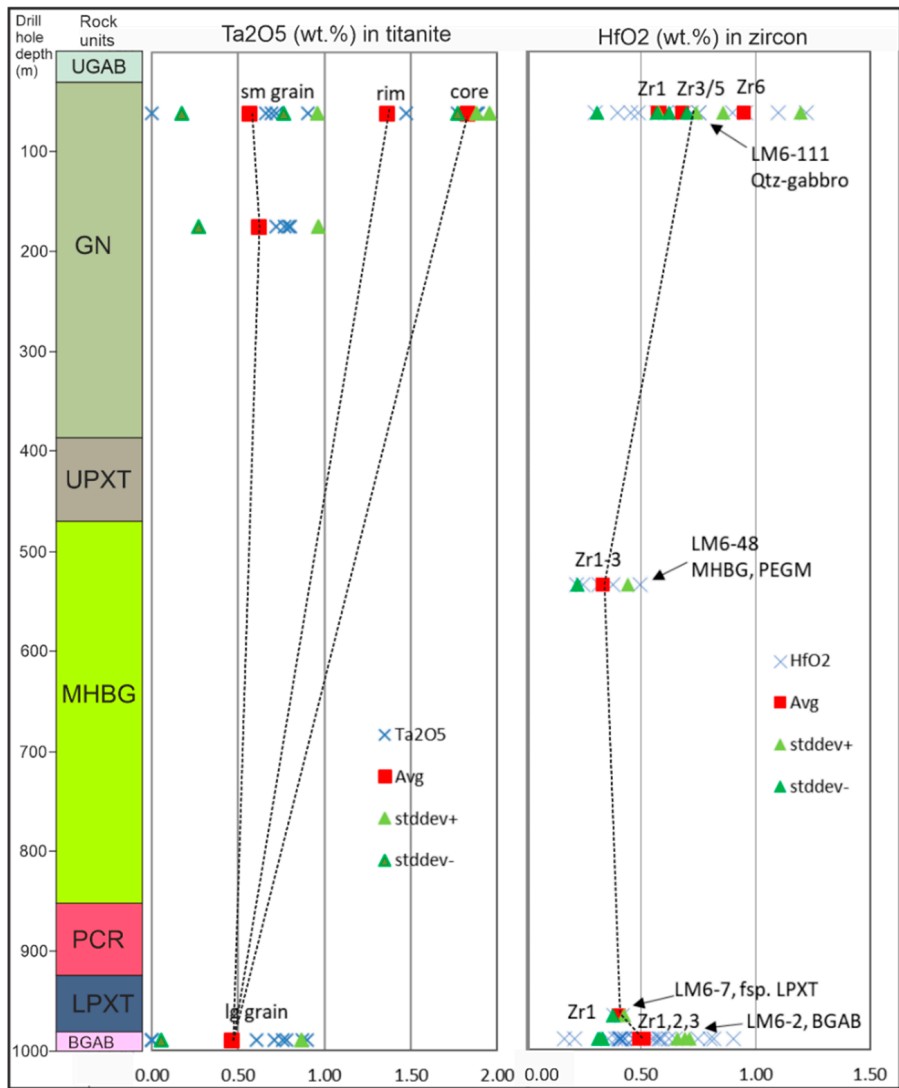

**Figure 11.** Trace elements in titanite, zircon (EPMA). Abbreviations: Rock units and types as in Figure 5. X $Ta_2O_5$ / $HfO_2$: single measurements, Avg: average value, stddev+/−: average plus/minus standard deviation, sm: small, lg: large, Zr1: zircon grain no.1, Qtz: quartz, fsp: feldspathic, Pegma: pegmatitic.

Chromite compositions from the Main Harzburgite unit in borehole LM-6 show in parts higher Cr- and higher Al-contents in the ternary $Cr^{3+}$-$Al^{3+}$-$Fe^{3+}$ diagram (calculated after [33]) than the chromites of the Main Harzburgite on the farms SH and UK (Figure 12A).

Chromite compositions in the Main Harzburgite unit of LM-6 have higher Cr- and Al-contents and are more primitive compared to those in the Main Harzburgite of the farms SH and UK (Figure 12A). The average Mg#, Cr/Fe and Cr/Cr + Al) ratios of chromite show an increasing trend, whereas the Cr/Cr + Fe + Al) ratio and the $TiO_2$ content show a decreasing trend from the Chromitiferous Harzburgite unit through the Main Harzburgite unit upwards (Figure 12B).

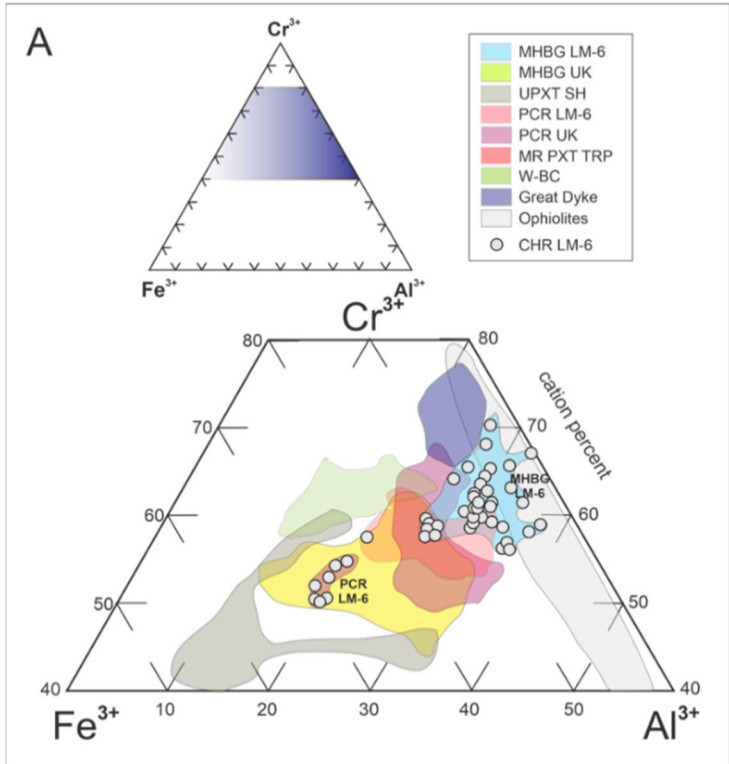

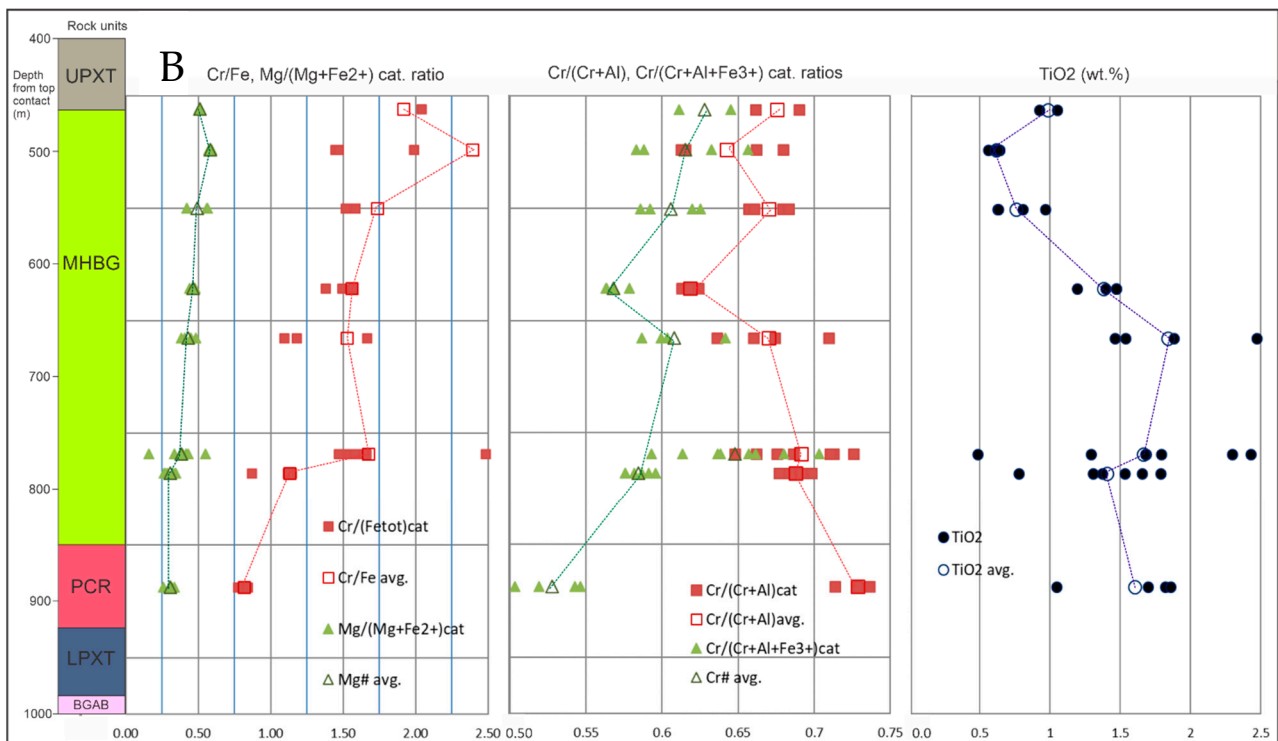

**Figure 12.** LM-6 Main Harzburgite and Chromitiferous Harzburgite chromite compositions in comparison with UK/SH and Bushveld Complex chromite data (**A**). Abbreviations of rock units as in Figure 5, MR PXT TRP: Chromites of Merensky Reef pyroxenite unit, Two Rivers Platinum Mine, W-BC: Merensky Reef unit, Western Bushveld. (**B**) average values and variation trends of Cr/Fe, Mg/(Mg + Fe$^{2+}$), Cr/(Cr + Al), Cr/(Cr + Al + Fe$^{3+}$) cation ratios and TiO$_2$ contents (wt.%) in chromite from Uitkomst chromitites within the LM-6 stratigraphic section.

The proportion of sulphide minerals is highest (max. estimated 13 vol%) in the Basal Gabbro unit, followed by the Lower Pyroxenite, Chromitiferous Harzburgite and lower MHGB units (on average 5, maximum 9 vol.%, Figure 7). Similar to UK and SH, pyrrhotite, chalcopyrite and pentlandite in the lowermost three rock units show variable value metal enrichment, with highest Co contents in Lower Pyroxenite and upper Main Harzburgite samples, and highest Ni content of pyrrhotite in the Chromitiferous Harzburgite (Figure 13a–d).

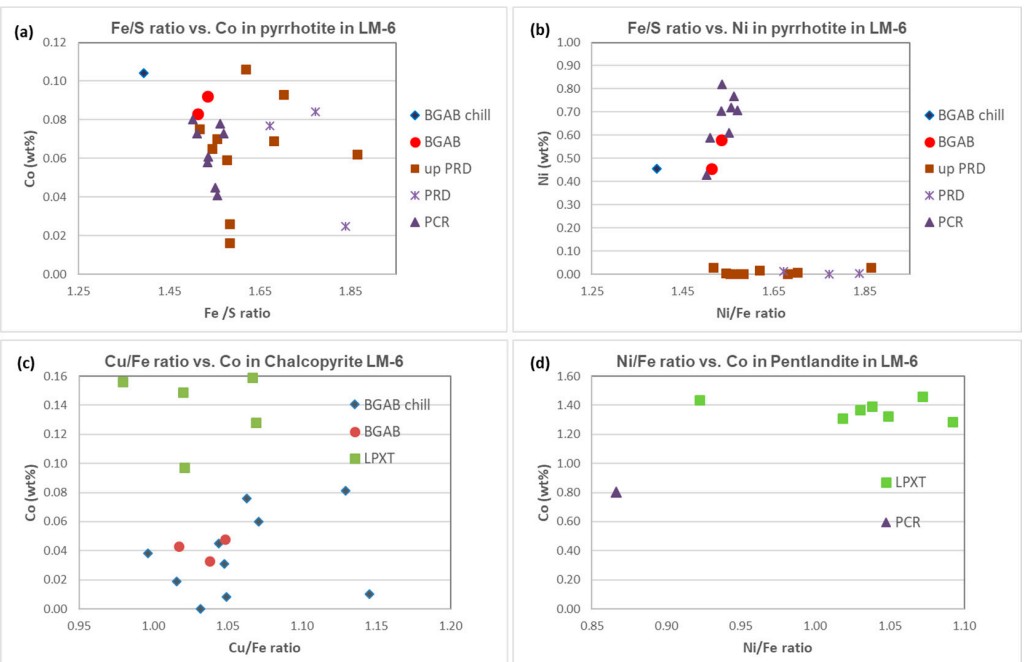

**Figure 13.** Pyrrhotite (**a,b**), chalcopyrite (**c**) and pentlandite (**d**) compositional data of the lower Uitkomst Complex in drill core LM-6.

### 3.3. Alteration and Contact Metamorphism

The degree of alteration in the LM-6 rocks at depth appears as intense as in the intersections further to the southeast showing constant intensity at least over 9 km distance towards northwest along the downdip extension.

The saussuritisation of An-rich plagioclase producing sericite, epidote and albite is common in the gabbroic units. This can be pursued further to the northwest of the intrusion and downdip at depth. The uralitisation of clino- and orthopyroxene to form hornblende, actinolite and tremolite, and chlorisation is common in the gabbroic and Lower Pyroxenite units (Appendix A, Figure A1, Plate 1d). Sections of the Lower Pyroxenite at LM show an increased clinopyroxene content, yielding lherzolitic and websteritic sub-rock types which are frequently amphibolitised. Serpentinisation of olivine and orthopyroxene is widespread in the ultramafic units, as is the formation of cube-shaped spinels in highly talc-carbonate altered Chromitiferous Harzburgite sections (Appendix A, Figure A2, Plate 2a). The ultramafic rocks in LM-6 produce a wide range of non-sulphide mineral assemblages.

Quartzite and dolomite assimilation by the magma recorded in the marginal zones, the Lower Pyroxenite and Gabbronorite units continues also with constant intensity along the downdip extension in LM-6. In areas with calc-silicate xenoliths, Ca-rich pyroxenes occur as frequent as orthopyroxenes or exceed their occurrence, and in areas with quartzite xenoliths in the Gabbronorite unit, quartz becomes the dominant intercumulus mineral.

A variety of secondary minerals after primary silicate, oxide and sulphide minerals occurs in the Chromitiferous Harzburgite unit. Least altered portions of primary silicates in its uppermost part at the transition to the Main Harzburgite unit, are serpentinised downwards, and in most areas characterized by talc-carbonate-chlorite formation after serpentine. Primary Ni- and Co-bearing pyrrhotite and pentlandite are altered and overgrown

by (arseno-)pyrite and hematite in the talc-carbonate altered parts of the unit (Appendix A, Figure A3, Plate 3c,d).

The metamorphic mineral content of the country rocks above and below the intrusion in drill core LM-6 compares favorably with the mineralogy the contact metamorphic aureole observed and analysed in field samples from the farm Uitkomst (Figure 2, Appendix A, Figure A1, Plate 1a,b). The vertical extent of the aureole of metapelitic to quartzitic rocks in LM-6 is comparable in terms of lateral width on the farms SH and UK.

### 3.4. Ore Mineralisation at Little Mamre

Trace element data of the primary sulphide minerals pyrrhotite, chalcopyrite and pentlandite (Figure 13) show that both the sulphide and nickel components in the ore may contain substantial proportions of the total Nickel and Cobalt budget mostly in the lower three rock units. In line with findings at SH and UK, whole rock Ni and Co contents in drillcore LM-6 increase upwards on average from 290 and 53 ppm (Basal Gabbro unit) to 2370 and 114 ppm (lower Main Harzburgite), Cu contents however decrease upwards from 855 ppm (Basal Gabbro unit) to 214 ppm (lower Main Harzburgite). Chemical assays of samples of the Chromitiferous Harzburgite and the lower Main Harzburgite units confirm upwards increasing trends for Ni and Co contents, but decreasing Cu, Pt, Pd, and Au contents (from 0.62 to 0.25 ppm for the latter three) with stratigraphic height (data ARM, 2014).

Similar to the silicates, the sulphide minerals are in parts altered during the late stage serpentinisation process: Pentlandite is altered to violarite (Appendix A, Figure A3, Plate 3d), occurring simultaneously with millerite, mackinawite, heazlewoodite and awaruite [3]. In copper-rich sulphides, native Cu can be formed. Secondary sulphides are further intergrown with oxide, sulphide and silicate minerals, thereby deteriorating separation and increasing metal extraction costs [34]. Secondary magnetite was precipitated as veins in between primary sulphides, and hydrated silicates grew into the sulphide mineralization (Appendix A, Figure A3, Plate 3b,c). Late stage galena and sphalerite occur subordinately within the primary sulphides.

Base metal sulphide ratios are very variable in serpentinites, indicated by for olivine corrected Ni/Cu ratios (Figure 8a). Chromites frequently show ferritised crystal rims, titano-magnetite and ilmenite in gabbroic rocks are partly altered to leucoxene and hematite, or magnetite is remobilized and precipitated.

### 3.5. Zircon Dating

Zircon grains were separated from pegmatoidal Lower Pyroxenite orthocumulate of the Uitkomst Complex at LM. The grains mostly represent fragments of bigger elongated crystals broken during sample crushing. Most of the fragments reveal oscillatory or banded zoning in CL images, and many of them contain re-crystallised melt inclusions dominated by quartz (Figure 14b). Many crystals are heavily fractured. Zircon grains were dated by U–Pb techniques; zircons were brittle, slightly altered and contained melt inclusions which could contain additional Pb. These have been avoided. Twenty-nine laser spot analyses were obtained from most pristine zircon domains apparently free of fractures and inclusions. Most analyses define a Discordia line with upper and lower intercept ages at 2054.0 ± 9.6 Ma and 564 ± 72 Ma, respectively (MSWD = 1.3, Probability 0.2, 2 sigma, *n* = 24), and the most concordant analyses a Concordia age of 2055.0 ± 5.3 Ma (MSWD = 0.44, Probability = 0.62, *n* = 12), which is interpreted to date magmatic zircon growth (Figure 14a).

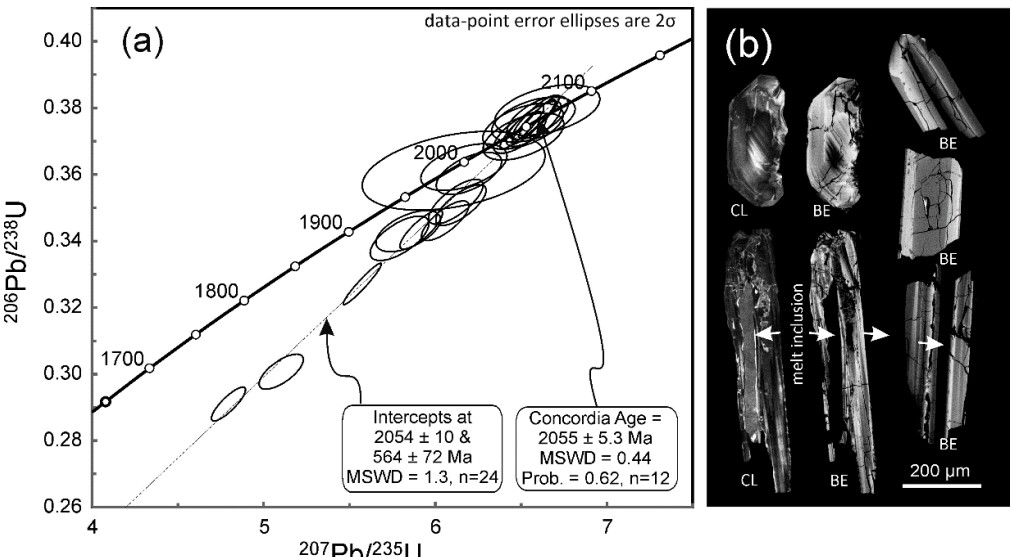

**Figure 14.** Results of zircon U–Pb dating presented in Concordia diagram (**a**), and of zircon imaging (**b**). Dated zircon grains commonly are fragments of elongated crystals showing oscillatory or banded zoning. Some crystals contain elongated melt inclusions, and many grains are heavily fractured. CL—cathodoluminescence; BSE—back scattered electron images.

## 4. Discussion

### 4.1. Mineralogy and Geochemistry Reflecting Magmatic System Conditions

Contrary to earlier assumptions, the intrusion dips with similar angle to the country rock (Figure 4), or rather the floor contact seems to undulate, showing a thick Black Reef quartzite underneath. The drill line on farm LM (Figure 6) implies the presence of dolomite to the southwest and the possibility of magma-country rock interaction.

In the LM part of the intrusion, magmatic system conditions are probably similar to the rest of the complex: A thin bottom chill zone of aphanitic fine-grained ophitic, altered and veined basal gabbro of the complex is overlain by at least 3 m of phaneritic, partly mineralised gabbroic orthocumulate, indicating a first pulse of basaltic composition along a shear zone. The Basal Gabbro aphanitic and phaneritic rock composition indicates that the chilled margins are more primitive at Uitkomst, and the phaneritic rocks are more evolved at SH and LM. It is not clear, at this stage, whether this is a function of the relatively low sampling density of the phaneritic rocks at SH and the chills at UK. Although the chill and phaneritic Basal Gabbro composition at UK are more primitive than the chill and the phaneritic rocks at SH and LM, the evolution trend continues only mildly from SH to LM, indicating that it could be a function of different degree of floor rock contamination (Figure 9a). Among the Upper Gabbro chill zones at LM, the LM-6 composition is the most primitive, possibly indicating its central position on top of the magma conduit (Figure 9b).

A strongly altered transition zone of gabbro into (clino-)pyroxenite to amphibolite with interstitial plagioclase indicates the beginning of the integration stage of peridotitic to komatiitic melt emplacement with sulphide and Fe-oxide melt exsolution. The transition of the Lower Pyroxenite to the Chromitiferous Harzburgite unit is indicated by an entirely talc-carbonatised and phlogopitised, fine-medium-grained, poikilitic adcumulate of olivine and orthopyroxene. For the modal mineralogy (Figure 7) alteration minerals have been attributed to the respective primary minerals (e.g., serpentine to olivine and orthopyroxene, amphibole to ortho- and clinopyroxene, saussurite to plagioclase, etc.).

The upwards distinctly rising Mg#, Cr/V and Ni/Cu ratios, and a strong decrease in Zr, Ba, and $TiO_2$ content (Figure 8) at the transition from Basal Gabbro unit into Lower Pyroxenite indicate the onset of ultramafic magma inflow. Geochemically this is characterised by the significant increase of Mg# from the Lower Pyroxenite (minimum 69.9) to the Chromitiferous Harzburgite unit (max. 92), and of Cr/V (2.2 to max. 596) and Ni/Cu

ratios (2.0 to 1000). Further decreases in Zr, Ba and $TiO_2$ content (Figure 8b) indicate the continuous inflow of ultramafic magma surges.

Compositional and differentiation variations are also responsible for the olivine-corrected Ni/Cu ratio, which is relatively high (up to 20) for the pyroxenitic to peridotitic units and up to two magnitudes lower for the gabbro(norit)ic units (above 2 on average).

The relative consistency of the above ratios and elements throughout the Chromitiferous Harzburgite und Main Harzburgite units indicates replenishment with magma pulses of similar primitive composition. A standstill of replenishment and the beginning of closed-system conditions the upper Main Harzburgite caused a gradational shift of dunite and harzburgite to the orthopyroxenite of the Upper Pyroxenite unit (Figures 7 and 8) and geochemical reversals [3]. Magma differentiation started under closed-system conditions with the first appearance of cumulus plagioclase in gabbronorite and reached its peak in quartz-rich diorite in the centre of the Gabbronorite unit. The Gabbronorite and Upper Gabbro units are geochemically evolved and strongly contaminated with quartz by assimilation of country rocks (Figure 3). The mineral assemblage is indicative of pegmatitic to hydrothermal formations from a $SiO_2$- and $H_2O$-rich, differentiated late-stage melt. The Upper Gabbro chill zone at LM is overall geochemically similar to the bottom chill zone, indicating the roof crystallisation of earlier magma flows with a trace element content and ratios close to B1 composition (Figure 9b); in particular the Cr content (317–570 ppm) reaches the magnitude of B1 sills. In the downdip part of the intrusion at LM, magmatic system conditions are probably similar to the rest of the complex.

Zircon saturation temperatures calculated for the observed major-element data and Zr concentrations after [35,36] at an assumed magma temperature of 1050 °C (based on 2-pyroxene magma temperature in [3]), showed an increase of that in the Basal Gabbro unit chilled margin from UK to LM (665 °C to 686 °C), and similar temperatures in the aphanitic Basal Gabbro and Upper Gabbro chill zone (699 °C, respective 727 °C) in LM-6.

The more primitive chromite compositions (higher Cr- and Al-contents) in the Main Harzburgite unit in borehole LM-6 compared to those in the Main Harzburgite of the farms SH and UK (Figure 12A) may indicate a more pristine composition closer to the source and less equilibration with a more evolved melt.

The finding of overall increasing Mg#, Cr/Fe, Cr/(Cr + Al) ratios and decreasing $TiO_2$ content of chromite with stratigraphic height (Figure 12B) is in line with [22], who interpreted the reverse compositional zoning in the Uitkomst chromitites as an indication of crystallisation in a magmatic conduit. Albeit part of the Uitkomst chromites lies within the field of the chemically most primitive compositions found in layered intrusions such as the Lower Zone of the Bushveld Complex (Figure 12A), the proposed conduit model for chromitite formation has much in common with models conventionally proposed for the formation of podiform and komatiite-hosted chromitites [22]. Comparison of LM-6 data with the whole rock geochemistry of drill core SH176 drilled on Slaaihoek in SE and described in [14,15,22] indicates a very similar course of the MgO, Mg#, $TiO_2$, Rb and Zr distribution of the two drill cores with stratigraphy. Compared to chromites analysed in the lower Chromitiferous Harzburgite unit in drill core SHM022 [22], the $TiO_2$-concentration of chromites analysed in the Chromitiferous Harzburgite and the Main Harzburgite unit in LM-6 is decreasing, and the Cr- and Mg-numbers are also increasing upwards; however, only few chromites have been analysed. Chromite compositional ratios such as Cr/Fe, Mg/(Mg + $Fe^{2+}$), Cr/(Cr + Al) and Cr/(Cr + Al + $Fe^{3+}$) from the Main Harzburgite unit in borehole LM-6 largely follow those of chromites of the Main Harzburgite unit on the farms SH (drill core SH-30, -31) and UK (drill core UD-17) [3]. This may indicate very similar processes and magmatic system conditions from the downdip side on farm LM to SH, such as country rock contamination in the Basal Gabbro and Lower Pyroxenite units, magma replenishment in the central Chromitiferous and Main Harzburgite units, as well as crystallisation from residual magmas to the ultramafic rocks in the Upper Pyroxenite, Gabbronorite and Upper Gabbro units. The Main Harzburgite and Gabbronorite units however are thicker at LM than at SH.

Apatite compositional heterogeneity (Figure 10) within samples and zonation in crystals may reveal a complex history of late-stage magma evolution at Uitkomst possibly within short timescales. Late-stage apatite is commonly associated with interstitial quartz in both the lower as well the upper gabbro. The paragenesis with mica and amphiboles is indicative of pegmatitic to hydrothermal formations under medium grade metamorphic conditions in a late-stage magma. The preservation of heterogeneous populations of apatite and of internally heterogeneous crystals requires short timescales (days to years) for these magmatic processes to occur [37].

The trace element chemistry of titanite in evolved amphibole-bearing gabbroic rocks and their metasomatised equivalents constitutes a tool for understanding late-stage igneous and metasomatic processes. Metasomatic and late-stage magmatic titanite grains have widely varying REE concentrations that likely relate to temperature, pressure, and local compositional effects [38]. The Uitkomst titanites in LM-6 stand in textural relationship with adjacent plagioclase, Fe-Ti oxides and mafic minerals and contain Al, Fe, Ta, Nb and some REE (Ce, Nd), and some crystals contain Mn, Mg, Na, Zr and V. Showing significant variability both between and within samples it is suggested that the titanite formed under metasomatic conditions. According to [38], titanite grains from altered olivine-bearing gabbro, troctolite microgabbro and metagabbro yielded lower Zr-in-titanite temperatures than Ti-in-zircon temperatures from coexisting zircon (i.e., generally below 750 °C), implying that these grains formed after zircon crystallisation with textural relationships that indicate metasomatic conditions.

In summary, magmatic system conditions prevailing in the Uitkomst Complex at LM indicate a mafic precursor intrusion forming the Basal Gabbro unit, followed by an integration stage of ultramafic magma interacting with country rock dolomite forming the strongly mineralised und altered Lower Pyroxenite unit. This is overlain by an open system of replenished ultramafic melt forming the Chromitiferous Harzburgite and Main Harzburgite units in a conduit-stage. The lack of massive chromite mineralisation in LM-6 may have its cause in permanent magma flows not allowing the precipitated chromite crystals to settle. Geochemical reversals in LM-6 between 400 and 500 m depth (Figure 8) indicate that the conduit stage came to an end when the flow of magma ceased and closed-system conditions prevailed, yielding olivine–chromite-dominated, orthopyroxene-dominated and plagioclase-dominated cumulates (Figure 7), respectively, with increasing height. The upper units reflect a magmatic differentiation sequence of the Upper Pyroxenite, Gabbronorite and Upper Gabbro units, where the last melt of dioritic composition is found in the centre of the Gabbronorite unit. Evolved late-stage melts formed also in pegmatoidal sections of the upper Main Harzburgite unit with accessory apatite, zircon and titanite. The composition of the upper chill zone may imply that it is a product of earlier roof crystallisation from a more primitive magma.

### 4.2. Alteration and Contact Metamorphism

Since alteration in the LM-6 rocks at depth is as intense as in the intersections further to the southeast, it is most likely that alteration during cooling of the igneous rocks has a deuteric character.

Based on their O-, H isotope study, [20] suggest selective, connate to meteoric fluid ingress, controlled by contact metamorphism and structural fluid pathways as alteration mechanism. These were identified among others by [13,39], leading to significant deuteric alteration at a stage in its cooling history. Metamorphic dehydration of muscovite, chlorite, clay minerals in shale is suggested as possible source for the external water needed to be included in magma [20]. This requires an enhanced heat flow situation for the intrusion ensured by a multiply replenished magma conduit situation and confirmed by the relatively large contact aureole around the Uitkomst complex [4] (Figure 2). The mineral content of the Uitkomst contact rocks above and below the intrusion in drill core LM-6 confirm pyroxene- to amphibole-hornfels facies metamorphism. Future research could address an

exact thermometric and heat flow characterisation of the contact aureole of the metapelitic and -carbonatic rocks.

A further fluid source for carbonation of part of the igneous rocks comes from the interaction of the magma forming the lower complex with dolomitic rocks (Figure 3) [19], causing decarbonation and release of $CO_2$-rich fluids by degassing. This increases the $O_2$-fugacitiy of magma on top of the lower intrusive "trough" scoured into the basal country rocks, creates favourable stability conditions for metal oxides, and triggers chromite and Fe-oxide precipitation. Furthermore, assimilation of Ca and Mg-carbonate from dolomitic rocks by the hot Mg-normative magma of the Lower Pyroxenite unit supports clinopyroxene formation (Figure 7), yielding lherzolitic and websteritic sub-rock types which are frequently amphibolitised. The continuation of the alteration mineralogy along complex up to borehole LM-6 indicates that the alteration conditions probably prevail on the entire downdip section of the complex. Based on the new findings, the country rock contamination in the marginal zones similarly continues with constant intensity at least over 9 kms distance towards north-west along the downdip extension.

Petrographic and textural evidence suggests that country rock assimilation of quartzite and dolomite also triggered the exsolution of oxide and sulphide melt forming respective mineralisations. In close succession dehydration and degassing processes introduced compositionally variable fluids, causing a multistage deuteric alteration process of silicates, oxide and sulphide minerals and forming a variety of secondary minerals (Appendix A: Figure A1, Plate 1d; Figure A2, Plate 2a,b).

Serpentinisation of olivine and orthopyroxene in the peridotite units causes reduced fluids, and $CO_2$-rich fluids form talc-carbonate altered rock sections of the Chromitiferous Harzburgite in a second auto-metamorphic phase of pervasive alteration of the ultramafic rocks [40]. The talc-carbonate-serpentine-chlorite dominance in the Chromitiferous Harzburgite unit, shows less altered portions with primary silicates at the transition to the main peridotite. $CO_2$-rich fluids from the degassing of dolomite xenoliths in the magma contributed to talcification-carbonation of the unit under the seal of the MCHR. Late-stage oxidised fluids associated with meteoric fluid ingress in the Chromitiferous Harzburgite unit gave rise to Ni-sulphide minerals coexisting with (arseno-)pyrite and hematite (Appendix A, Figure A3, Plate 3c,d). Future research could address an exact thermocompositional characterisation of the alteration stages of the Chromitiferous Harzburgite unit.

In LM-6 fluids produced a wide range of non-sulphide assemblages, despite the relatively restricted compositional range within each rock type. The uralitisation and chloritisation of clino- and orthopyroxene to form hornblende, actinolite, tremolite, and chlorite is common in the gabbroic and Lower Pyroxenite units, whereas the saussuritisation of An-rich plagioclase producing sericite, epidote and albite is common in the gabbroic units. This can be pursued further to the northwest of the intrusion and downdip at depth.

### 4.3. Mineralisation

At a cooling stage of temperatures below 600 °C, subsolidus processes such as the exsolution of a Cu-rich intermediate solid solution (ISS) forming chalcopyrite and cubanite from a monosulphide solid solution (MSS) takes place, and the exsolution of pentlandite flames from MSS pyrrhotite at around 605 °C, as displayed by the sulphide mineral assemblage (Appendix A: Figure A3, Plate 3a). The amount of Nickel and Cobalt in the sulphide minerals (Figure 13) is controlled by the degree of interaction of exsolved sulphide liquid with larger volumes of replenished silicate melt and its opportunity to separate [14], which is highest in the central part of the complex in the conduit stage.

The alteration of pentlandite to millerite, mackinawite, violarite (Appendix A: Figure A3, Plate 3d), heazlewoodite and awaruite, and the formation of native Cu in copper-rich sulphides indicate the low temperature and partly reducing conditions of the hydrothermal fluid forming the secondary sulphide minerals. Hydration of olivine and orthopyroxene

under greenschist facies conditions gave rise to serpentinites, hosting in some cases Ni-sulphide assemblages rich in violarite, mackinawite and millerite.

During deuteric alteration hydrothermal fluids caused the growth of hydrated silicates into the pentlandite-chalcopyrite-pyrrhotite mineralisation, and secondary magnetite was remobilised as veins into primary sulphides. Secondary sulphides caused further intergrowth of oxide and sulphide minerals and deteriorated separation and increased its costs [34].

Hydrothermal fluids are probably responsible for the subordinate occurrence of (late stage) galena and sphalerite, Zn and Pb exsolved from primary sulphides or introduced by metamorphic fluids from the basement and sulphidised shales.

Sprouting of (arseno-)pyrite (Appendix A: Figure A3, Plate 3c) and cobaltite [9] on primary and secondary sulphides indicates the remobilisation of Fe, As and Co by S-rich fluids at a late hydrothermal stage.

Primary chromite, titano-magnetite and ilmenite show alteration under the influence of oxidised low-T fluids, leading to the ferritisation of chromite crystal rims, and formation of leucoxene, secondary magnetite and hematite. Ferritisation of chromite crystals by oxidised low-T fluids decreases the Cr-content of spinels [41,42]. Later oxidation processes of Ti-magnetite may mobilise Ti and cause rutile and sphene formation in pegmatoidal sections and in the Gabbronorite unit.

Alteration has a considerable effect on mineral processing and beneficiation in that a coat of serpentine and talc on sulphide minerals negatively influences flotation behaviour of the sulphides [43], whereas grain size coarsening of value metal sulphide crystals is beneficial to liberation. Secondary pyrite crystallisation due to remobilising S-rich fluids is also detrimental for the flotation of Nickel sulphide-bearing particles because it dilutes the Nickel concentrate.

### 4.4. Age of the Uitkomst Complex

Results of U–Pb dating, and quartz-rich re-crystallised melt inclusions in zircon provide evidence that zircon in the Uitkomst Complex crystallised from highly fractionated intercumulus melts at 2055.0 $\pm$ 5.3 Ma, most likely at temperatures between 940 °C and 670 °C. The last interpretation is in good agreement with results of two studies on host zircons and melt inclusions systematically carried out by [44,45] on zircon grains from different mafic rocks of the Rustenburg Layered Suite. Our new Concordia age of 2055.0 $\pm$ 5.3 Ma overlaps within error with a $^{207}$Pb/$^{206}$Pb TIMS age of 2054 $\pm$ 7 Ma obtained from baddeleyite crystals by [23], and a weighted mean $^{207}$Pb/$^{206}$Pb CA-ID-TIMS age of 2057.64 $\pm$ 0.69 Ma derived from zircon by [15] from the main body of the Uitkomst Complex. All results together indicate that all units of the Uitkomst Complex, including the NW-dipping part investigated during this study, crystallised within the same period from intercumulus melts. The data also overlap those obtained by CA-ID-TIMS from several units of the Rustenburg layered suite [30,45], suggesting that the Uitkomst Complex formed part of a plumbing system, which fed the Bushveld Complex magma chamber. Due to high chromite content of the Uitkomst Complex [10], it was most likely formed together with units of the Lower Critical Zone [22].

### 5. Conclusions

- The first complete intersection of the complex at LM supports lithological and mineralogical continuity and an increase in thickness of the Main Harzburgite and Gabbronorite units in the downdip extension compared to the intersections on UK and SH (e.g., drillhole SH176). The Chromitiferous Harzburgite unit contains less and thinner massive chromitite layers, whereas the Main Harzburgite unit displays layers of pegmatoidal rocks in its central and upper parts.
- The shallowly dipping trough to sill-like intrusive body continues bedding parallel towards northwest without signs of plunging into a subvertical feeder channel. Mineralogical and lithogeochemical trends in LM-6 are similar to previous intersections;

their consistency in the MHZG unit supports a model that magma surges replenished the conduit with magma of almost constant composition. Zircon saturation temperatures calculated for the observed major-element data and Zr concentrations showed a minor increase for the Basal Gabbro chilled margin from UK to LM, but a significant decrease from the margins to the centre.

- Chromite compositions of the Main Harzburgite unit at LM are more Cr- and Al-rich than at UK and chemically less evolved than Bushveld Complex Merensky Reef chromites, reflecting the deeper part of the conduit. The upwards increasing Mg#, Cr/Fe and Cl/(Cr + Al) and decreasing $TiO_2$ trends may indicate a formation by magma replenishment similar to formation mechanisms suggested for podiform and komatiite-hosted chromitites.
- Late-magmatic apatite, titanite and zircon in pegmatoidal portions of ultramafic and in mafic rocks originated from an evolved, $H_2O$-rich late-stage melt as part of the differentiation sequence. Apatite compositional heterogeneity reveals a complex history of magma mixing with at least two components and crystallisation over short timescales. Carbonatic fluids probably contributed to the formation of titanite from the Ti-bearing melt and magmatic ilmenite.
- "Autometamorphic" fluids derived from contact metamorphism of pelitic and carbonate rocks caused pervasive hydrothermal (deuteric) alteration such as serpentinsation, talc-carbonation, uralitisation and saussuritisation, affecting both mafic and ultramafic units. S-bearing fluids produced secondary pyrite overgrowth, which deteriorates flotation results. Common coatings of talc and mica particles on sulphide minerals have also an opposing effect on the flotation process.
- Ore mineral remobilisaton and alteration, as well as hydrated silicate growth into sulphides promotes multiple mineral intergrowth and decrease of grain size diameters, negatively affecting pentlandite and chalcopyrite liberation.
- U–Pb dating of zircon from the Lower Pyroxenite Unit indicates intercumulus melt crystallisation within the Uitkomst Complex at 2055.0 ± 5.3 Ma, coeval to the Bushveld Complex, suggesting that mafic rocks of both complexes form part of the same magmatic event.
- The intrusion of ultramafic magma into carbonate rocks is considered to represent a highly prospective environment for the formation of magmatic Ni-Cu-PGE-Cr deposits. Ideal situations are host-rock-concordant feeder channels along the Mpumalanga escarpment at the elevation of the Malmani dolomite.

**Supplementary Materials:** The following are available online at https://www.mdpi.com/article/10.3390/min12010022/s1, Table S1: Results of U–Pb zircon dating. Table S2; Chromite (sheet 1), apatite (sheet 2), titanite (sheet 3) and zircon (sheet 4) compositional data (EPMA) of LM sections. Table S3: Modal primary and secondary mineral proportions in sections of drill core LM-6 Nkomati Nickel, Uitkomst Complex, Little Mamre 538.

**Author Contributions:** Conceptualization, C.G.; methodology, C.G. and A.Z.; software, C.G.; validation, C.G.; formal analysis, C.G.; investigation, C.G. and A.Z.; resources, C.G.; data curation, C.G.; writing—original draft preparation, C.G.; writing—review and editing, C.G. and A.Z.; visualization, C.G. and A.Z.; supervision, C.G.; project administration, C.G.; funding acquisition, C.G. All authors have read and agreed to the published version of the manuscript.

**Funding:** This research was funded by the National Research Foundation of South Africa, funding for Rated Researchers IFR170126218646 to CG: https://www.nrf.ac.za/funding, and by an incentive fund of the University of the Free State, Bloemfontein, South Africa.

**Acknowledgments:** The authors are grateful to Mark Davidson, ARM-Nkomati Nickel Mine: Donation of drill core material for research, Gelu Costin, Dept. of Geology, Rhodes University, Electron Microprobe facility: EPMA investigation. We acknowledge support by the Department of Geology, Univ. of the Free State, Bloemfontein, South Africa, for access to XRF, SEM-EDX analytical facilities. The preparator D. Radikgoma is thanked for polished sections. M. Yudovskaya, Department Geology, University of the Witwatersrand, is acknowledged for a drill core log.

**Conflicts of Interest:** The authors declare no conflict of interest.

## Appendix A

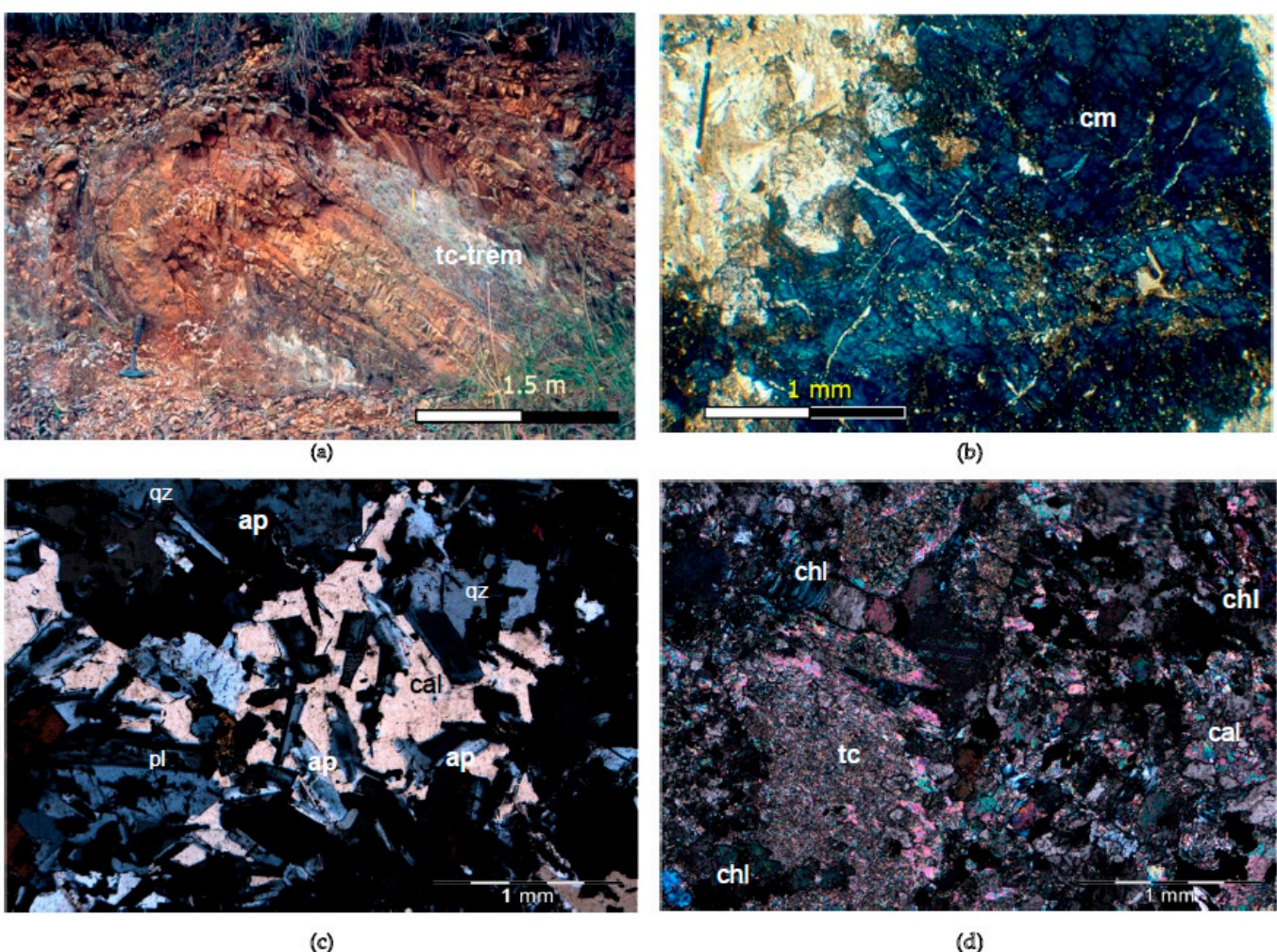

**Figure A1.** Plate 1. Country rocks and lower intrusive units: (**a**) Talc-tremolith-rich (tc-trem) layers in chert-rich dolomite (Oak Tree Formation, Malmani Subgroup) on farm Uitkomst; (**b**) Micropho-tograph (MP) (crossed polarisors, XPL) of cm-sized Corundum (cm) porphyroblasts in meta-pelite contact aureole (max. 30 m width); (**c**) MP (XPL) of Basal Gabbro: Interstitial calcite (cal), quartz (qz), plagioclase (pl) and apatite (ap); (**d**) MP (XPL) of Lower Pyroxenite: Talc (tc)-carbonate-amphibole-chlorite (chl) alteration.

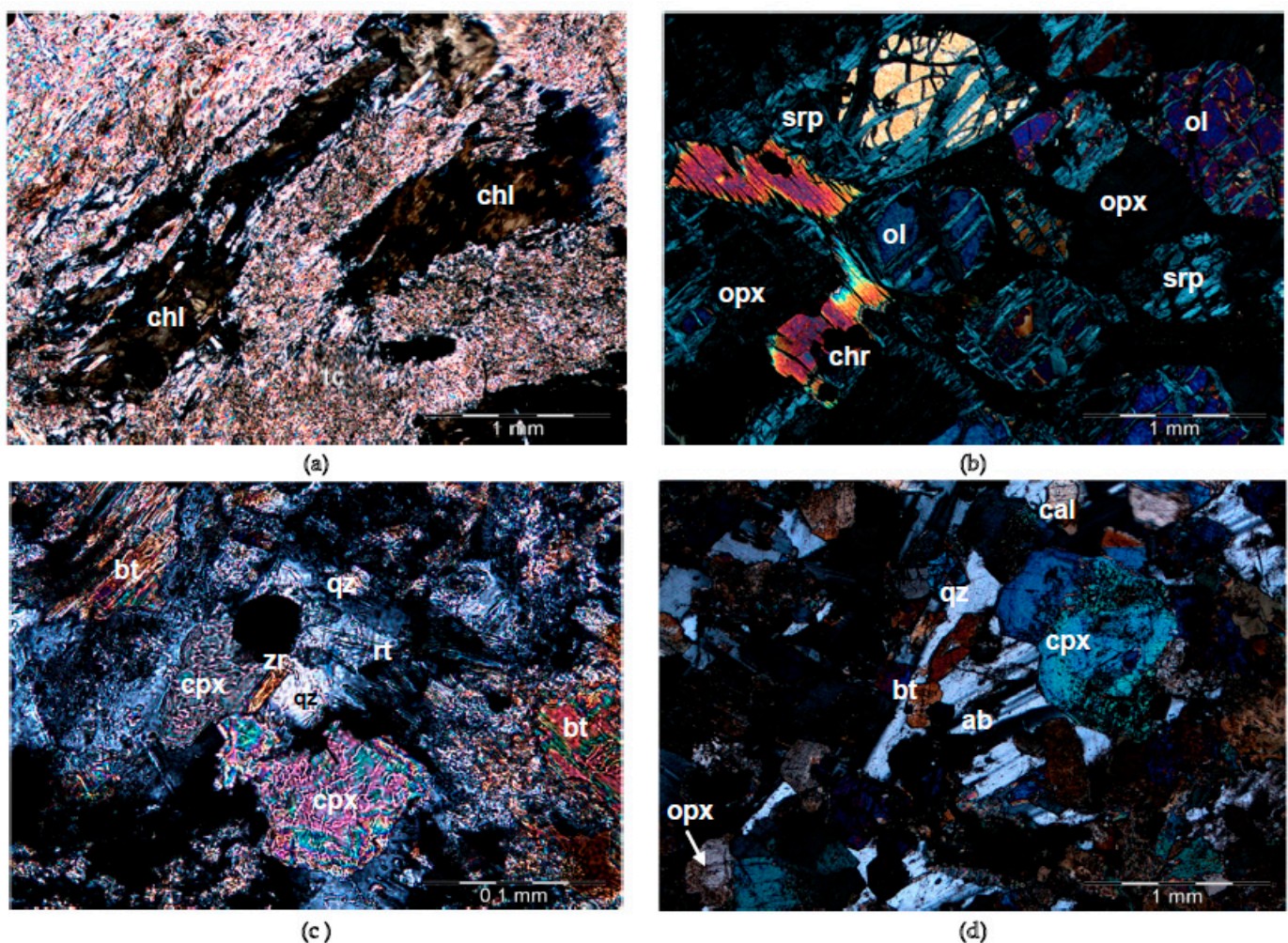

**Figure A2.** Plate 2. Central and upper intrusives: (**a**) MP (XPL) of talc-chlorite-rich sheared Chromitiferous Harzburgite; (**b**) MP (XPL) of Main Harzburgite serpentinized (srp) olivine (ol) in orthopyroxene (opx) matrix with interstitial magnetite and chromite (chr); (**c**) MP (XPL) of altered Gabbronorite amphibolitized clinopyroxene/orthopyroxene and biotite (bt) with interstitial quartz, albite, zircon (zr), rutile (rt); (**d**) MP (XPL) of Upper Gabbro clinopyroxene (cpx), plagioclase (pl), interstitial quartz (qz), and albite (ab).

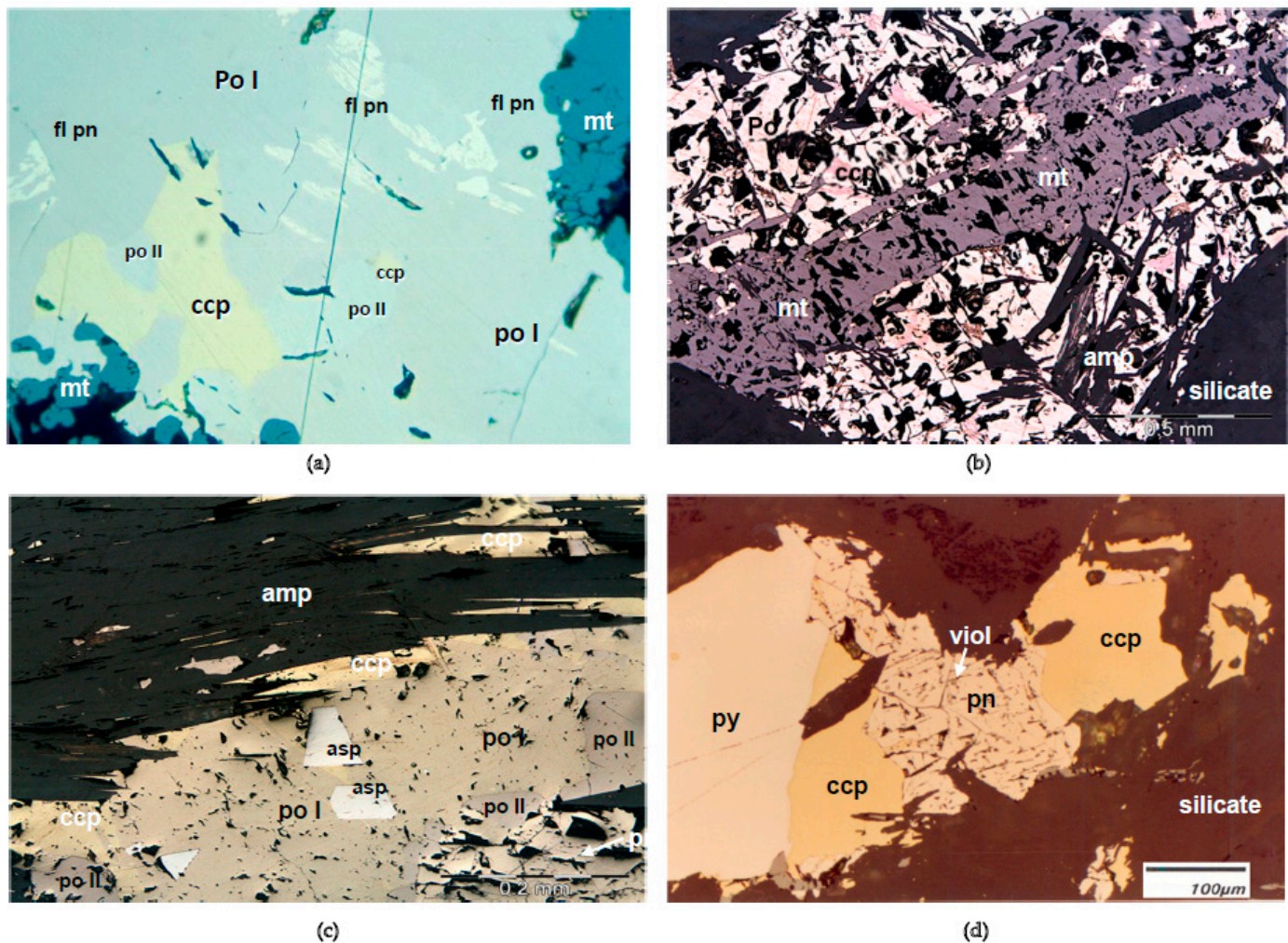

**Figure A3.** Plate 3. Ore mineralogy and textures in LM-6 and at UK: (**a**) MP in reflected light (RL) Aggregate of pyrrhotite (po), flames pentlandite (fl pn), chalcopyrite (ccp) and secondary magnetite (mt) in Lower Pyroxenite unit on UK; (**b**) MP (RL) of Po-pn-ccp-aggregate with secondary magnetite and amphibole (amph) in wehrlite of Lower Pyroxenite unit; (**c**) MP (RL) of Po-pn-ccp-aggregate with hypidiomorphic arsenopyrite (asp) and amphibole growth into sulphides of Chromiti-ferous Harzburigte unit; (**d**) MP (RL) of composite sulphide particles of pyrite (py), chalcopyrite and pentlandite, partially replaced by violarite (viol., darker patches within pn) on UK.

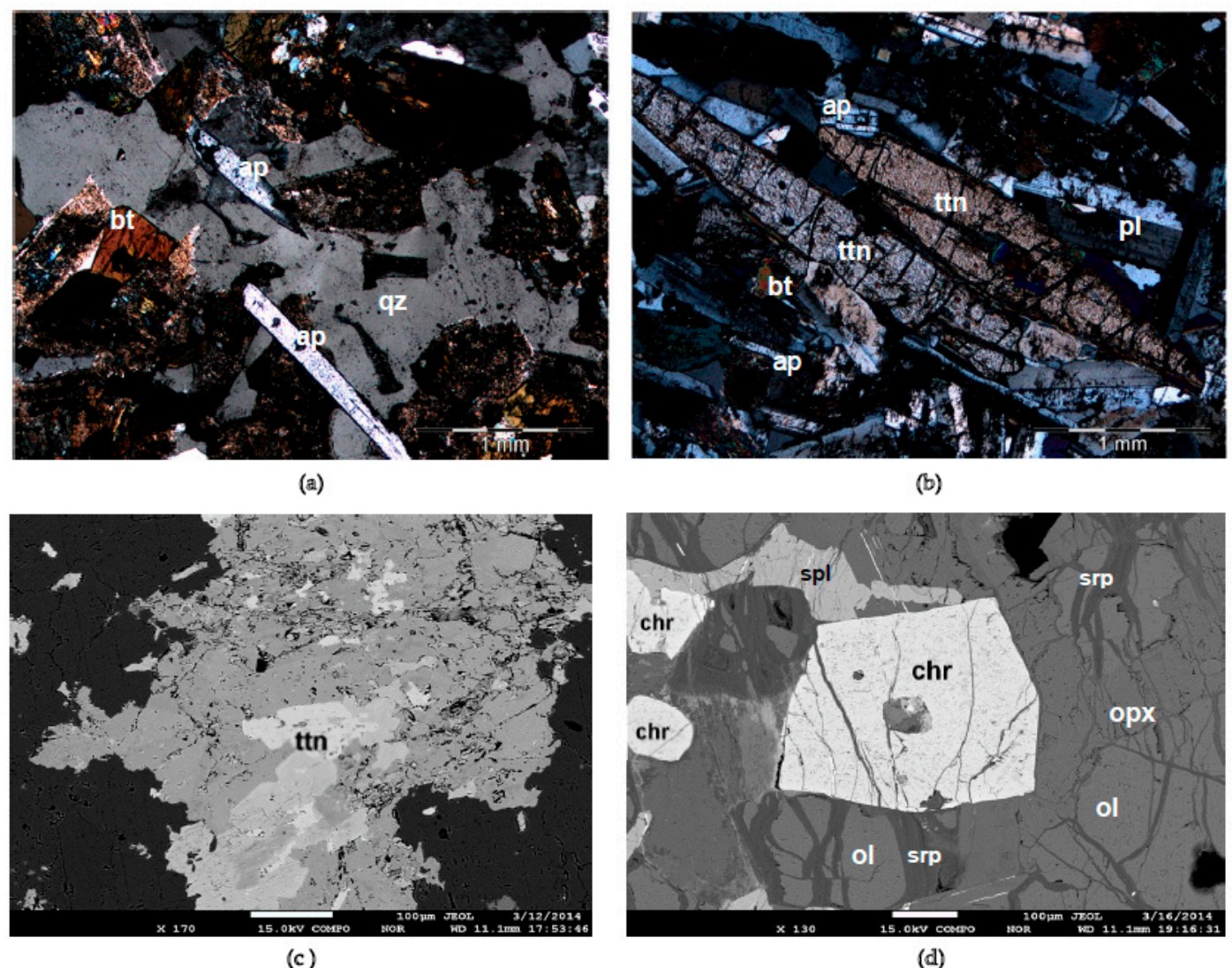

**Figure A4.** Plate 4. Late-stage minerals and harzburgite texture in LM-6: (**a**) MP (XPL) of apatite (ap) crystals in a quartz-gabbro of the Gabbronorite unit; (**b**) MP (XPL) of titanite (ttn) and apatite (ap) crystals in the Gabbronorite unit; (**c**) SEM-BSE image of titanite (ttn) aggregate in Gabbronorite unit; (**d**) SEM-BSE image of chromite (chr) crystal with melt inclusion and spinel (spl) in moderately serpentinized (srp) Main Harzburgite unit.

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
