# Peer review of "Downdip Development of the Ni-Cu-PGE-Bearing Mafic to Ultramafic Uitkomst Complex, Mpumalanga Province, South Africa"

_minerals, doi:10.3390/min12010022_

Round 1

Reviewer 1 Report

The manuscript is devoted to description of geology, chemistry and mineralogy of the Uitkomst Complex (UC), situated to the east from Bushveld intrusion. UC in the exposed parts is relatively good investigated (Gauret at al., 1995; Gauret, 2001; Li et al, 2002; Maier et al., 2004; Yudovskaya et al., 2016; et al.). The novelty of new materials are in the first description of core of the borehole LM-6, crossing the UC in not exposed area. Authors give detailed description of lithology, mineralogy, geochemistry and geochronology of the UC. These materials should be of interest for many geologists. Described features of the units of the UC as well the age of the UC mostly coincide to the published data, but it is not belittle author’s materials. In the same time, the manuscript seems to be not properly organized and needs major revision.  

Main remarks.

  1. Abstract is not informative. It mostly shows – what items were investigated, but there are almost no results of these investigations.

  1. It is not easy to read the manuscript because of many abbreviations (geography, names of geologic units, minerals). Any reader firstly has to learn their meanings. Better to use abbreviations only for geographic terms. For minerals would be better to use symbols recommended by Whitney and Evans (Am. Miner., 2010). In Suppl.Mat. you should explain any abbreviation and mostly not to use them.

  1. In introduction authors declare «The research question and aim of this study is to find out how the mineralogy, the whole rock and mineral chemistry of the UC, and the thickness of the lithological units continue in their downdip development.». If so, why they do not compare their data on LM-6 with investigated earlier boreholes SH176, SHM022, SHM202 and other boreholes, drilled in SE and well described in Li (2002), Maier (2004) and Yudovskaya (2016)?

  1. Authors discuss composition of minerals only of late magmatic or postmagmatic stage. It would be much more important to discuss features of main rock-forming minerals – Ol, Opx, Cpx, Pl and to place mineral analyses (not mean values, see below) to Suppl. Tables.

  1. Table 1 has no sense. It is similar to average temperature in a hospital including morgue. Better to place individual analyses in Supplementary materials. To place also apfu for minerals and to recalculate FeOtot to Fe2O3 and FeO (at least in apfu) for chromite. And exclude bad analyses with total higher than 101 Wt%. And again unknown abbreviations: PRD-PEGM, GABNOR.

Some other comments and questions.

Line 11-16. This statement is rather for introduction.

Line 39-40. Nickel. Not capital letter.

Line 43. What means MSB?

Fig 2. What means THS debris? What is the composition of Rooihoogt formation?

Fig. 3. In description: not stratigraphic but lithologic. There are shown many diabase sills and feeding dykes. In text they are not mentioned. Does it mean that in LM-6 younger diabase (see Yudovskaya et al., 2016) are absent? If so, please write it in the text. It is important to discuss the reason of relatively low-T secondary alteration.

Line 83 (and below). Why not [1, 3, 5-13 etc.] but [6,7] [1,8,9] and so on.

Why Fig. 3 and Fig. 6 have reverse directions? The first from NE to SW and the second from SW to NE.

Fig. 5. What is PXR, MHZBG, PCR, LXZBG? In Fig. 3 explanation only for MHBG.

Line 116. The statement contradict to Fig. 5. If magma pulsed ten to hundred times, why Mg# and Cr/V curves are so smooth?

Fig 7. Better to use abbreviations of minerals recommended by Whitney and Evans (Am. Miner., 2010). And more than 50% of Qtz in rocks doesn’t corresponds to norite nor diorite. What it is?  

Line 267. Accessory or secondary? There are different terms.

Line 278. What is PCR?

Line 312. Gabbronorite. Not capital letter.

Line 319. Contminated by quartz (mineral) or by SiO2?

Line 382-383. Amphibole, albite and biotite are not accessory minerals.

Line 392. Not stratigraphy, lithology.

Fig. 10. Lithology instead of stratigraphy.

Line 408-409. Declaration without any arguments.

Fig. 12. Never met Cr/Fetot ratio for chromite. What is its meaning? It is not discussed in the text.

Line 449. Lithologic instead of stratigraphic.

Line 477. Chlorization - chloritization

Line 482. What means auto-metamorphic?

Line 498. Nickel and Cobalt. Not capital letters.

Line 526. Discordia. Not capital letters.

Line 537. BE – back scattered. Use BSE.

Line 626-627. “Showing significant variability both between and within samples it is suggested that the titanite formed under metasomatic conditions.” Before it was written about late magmatic and metasomatic stage. What statement is correct? Metasomatic conditions might be much younger than UC cooling.

Line 703. Nickel and Cobalt. Not capital letters.

Line 704. exsolved sulphide liquid. There were no discussion about sulphide liquid.

Line 745. Rustenburg Layered Suite. Where it is situated?

Line 755. Lower Critical Zone. Where it is situated?

Line 816 (and below). Tremolith. Tremolite.  

Line 888. Not 2015, 2016.

Suppl –table A1. No explanation of abbreviations in line 3. I don’t understand meaning of “cplag fr”, “iplag fr” and some others.

In column “comment”. Many abbreviation without explanation. For what – “very good polish” and some other comments of the same style.

Author Response

Dear reviewer 1,

Thank you for taking the time to thoroughly go through the manuscript and giving valuable comments and criticism to improve its quality.

Although the downdip development of the complex under cover does not reveal spectacular new findings, it nevertheless shows variations to the southeast, i.e. variation in thickness of the rock units and the entire complex, modal mineral proportions of gabbronorite (more quartz and presence of titanite), and mineral chemistry of late stage minerals e.g. apatite, titanite and zircon.

The manuscript has been revised according to logic and restructured w.r.t. to several result passages been moved to the discussion session. Text passages (highlighted in yellow) and most figures have been revised.  The detailed comments and questions have all been addressed. Please see attached manuscript.

Regarding the main remarks:

  1. The abstract has to be short (250 words) which occupies space for what, where(for), how and by whom, so the results of investigation so far have been reported in general. The abstract has been revised. If there is more space, it can easily be elaborated with more detail. The details are given in the conclusions. Please indicate if it is necessary to expand results in the abstract.

  1. The rock unit abbreviations have been maintained because to write them out would take too much space. It can be replaced if regarded necessary. Please advise. In Suppl.Mat. abbreviations are explained.

  1. Comparison of LM-6 data with the whole rock geochemistry of borehole SH176 drilled on Slaaihoek in SE and well described in Maier et al. (2018) and Yudovskaya (2016) is carried out in the discussion chapter 4.

  1. Some main rock-forming minerals – Opx, Cpx, Pl have been analysed exemplarily but not systematically over the entire stratigraphy, so the focus will be the late stage the original analyses will be put to Suppl. Tables. This will take a bit of time due to full schedule of the senior author.

  1. Individual analyses in supplementary materials, apfu for minerals will be given and FeOtotto Fe2O3 and FeO (at least in apfu) for chromite will be recalculated. Bad analyses will be excluded. Abbreviations are explained. The atomic unit calculations for the digital appendix of mineral chemistry can only be performed in 8 days or be delivered afterwards, because of lack of access to literature. This will take a bit of time (until 28.11.2021) due to full schedule of the senior author.
  2. The MS with the chnaged passages (highlighted yellow) as corrected now has been attached below. 
  3. Kind Regards C. Gauert

Reviewer 2 Report

Please see coments in attached file.

I hope it will help to improve your manuscript.

Author Response

Dear reviewer 2,

Thank you for taking the time to thoroughly go through the manuscript and giving valuable comments and criticism to improve its quality.

Although the downdip development of the complex under cover does not reveal spectacular new findings, it nevertheless shows variations to the southeast, i.e. variation in thickness of the rock units and the entire complex, modal mineral proportions of gabbronorite (more quartz and presence of titanite), and mineral chemistry of late stage minerals e.g. apatite, titanite and zircon.

The manuscript has been revised according to logic and restructured w.r.t. to several result passages been moved to the discussion session. Text passages (highlighted in yellow) and most figures have been revised.  The detailed comments and questions have all been addressed. Please see attached manuscript.

All comments on the supplement materials figures A1 to A4 (field photo, microphotographs and BSE images) have been carried out but cannot be uploaded as second file on this site.

Kind regards, C. Gauert

Reviewer 3 Report

The manuscript: minerals-1441901 titled “Downdip development of the Ni-Cu-PGE-bearing mafic to ultramafic Uitkomst Complex, Mpumalanga Province, South Africa”, presents an interesting research study. The manuscript is well written, with Figures being correctly displayed and very informative. Results from petrographic escriptions, XRF, EPMA and LA-ICP-MS, as well as U–Pb analyses of zircons, provides the necessary support for the discussion section. Conclusions and Abstract are balanced, well written, and successfully display the major outlines of this paper. Therefore, I suggest that this paper should be published in its present form in the Journal of “Minerals”.

Author Response

Dear reviewer 3,

Thank you very much for taking the time to thoroughly go through the manuscript and trusting the quality of the paper.

Following some other comments the manuscript has been revised according to logic and restructured. Few text passages (highlighted in yellow) and most figures have been revised, please see attached manuscript.

We ask for your permission to publish in the attached form, however with some shift of the mineral chemistry tables to the digital appendix after cleaing and recaluculation of the afu for ap, chr, ttn and zr.

Kind regards,

C. Gauert

Round 2

Reviewer 1 Report

I refuse to review "corrected" version of the manuscript because it is
not properly corrected. Only ''cosmetic corrections''.
My main suppositions were ignored, see below.
1. It is not easy to read the manuscript because of many abbreviations
(geography, names of geologic units, minerals). Any reader firstly has
to learn their meanings. Better to use abbreviations only for geographic
terms. For minerals would be better to use symbols recommended by
Whitney and Evans (Am. Miner., 2010). In Suppl.Mat. you should explain
any abbreviation and mostly not to use them.
Author response: The rock unit abbreviations have been maintained
because to write them out would take too much space. It can be replaced
if regarded necessary. Please advise. In Suppl.Mat. abbreviations are
explained.
Till now I am sure that most of abbreviations in the text should be
normally explained. There are no restrictions in papers volume. And
abbreviations of minerals also differ from recommended (Whitney, Evans,
2010). Moreover, Suppl-table A1-modal mineralogy is the same as in the
first variant. But this table looks like raw material with unclear
abbreviations and vague comments. I think that to publish such kind of
tables will be bad for the image of the Journal.

2.In introduction authors declare «The research question and aim of this
study is to find out how the mineralogy, the whole rock and mineral
chemistry of the UC, and the thickness of the lithological units
continue in their downdip development.». If so, why they do not compare
their data on LM-6 with investigated earlier boreholes SH176, SHM022,
SHM202 and other boreholes, drilled in SE and well described in Li
(2002), Maier (2004) and Yudovskaya (2016)?
Author response: Comparison of LM-6 data with the whole rock
geochemistry of borehole SH176 drilled on Slaaihoek in SE and well
described in Maier et al. (2018) and Yudovskaya (2016) is carried out in
the discussion chapter 4.

I couldn't find discussion in Chapter 4.

3. Table 1 has no sense. It is similar to average temperature in a
hospital including morgue. Better to place individual analyses in
Supplementary materials. To place also apfu for minerals and to
recalculate FeOtot to Fe2O3 and FeO (at least in apfu) for chromite. And
exclude bad analyses with total higher than 101 Wt%. And again unknown
abbreviations: PRD-PEGM, GABNOR.

I couldn't find mineral analyses in Suppl. Mat.

I am ready to review the "polished" and ready for publishing manuscript
but not really corrected uploaded variant.

Best regards,

Author Response

Dear Reviewer,

thank you for your thorough criticism, it is very helpful. Apologies that we uploaded previous "corrected" version of the manuscript in order to meet deadline without implementing all changes requested.

We have addressed your concerns in the following way:

1. The many abbreviations in the manuscript have been avoided except for
the geographic names of the three farms Uitkomst (UK), Slaaihoek (SH) and Little Mamre (LM). All rock units and mineral names have been written out. For minerals in Supplement material Table A1 and A2 symbols recommended by
Whitney and Evans (Am. Miner., 2010) have been used.

Suppl-table A1-modal mineralogy has been overworked, abbreviations have been avoided and vague comments deleted. We would like to give data as basis of Figure 7.

2. We tried to compare the downdip mineralogy, the whole rock and mineral
chemistry of the complex and the thickness of the lithological units
on farm Little Mamre with that of the boreholes SH176 and SHM022, drilled in Slaaihoek/Uitkomst and described in Li (2002), Maier (2004) and Yudovskaya (2016), and Gauert (1998) in lines 634 to 650 in the discussion chapter 4.

3. Table 1 in the text has been deleted. Instead, individual analyses have been presented in in Supplementary materials, Table A2. Apfu for minerals have been calculated. FeOtot has been recalculated to Fe2O3 and FeO for chromite using the procedure by Barnes 2004. Bad analyses with total higher than 101 Wt%, have been excluded, unknown abbreviations avoided.

The changed passages in the MS are highlighted in yellow color.

We would appreciate if you could review the "polished" and ready for publishing manuscript again. 

Best regards,

C. Gauert

Round 3

Reviewer 1 Report

Now o'k